# Global Convergence Rate of Deep Equilibrium Models with General Activations

**Lan V. Truong**                                                 *lan.truong@essex.ac.uk*
*School of Mathematics, Statistics and Actuarial Science*
*University of Essex*

**Reviewed on OpenReview:** *https://openreview.net/forum?id=XPREcQlAMO*

## Abstract

In a recent paper, Ling et al. investigated the over-parametrized Deep Equilibrium Model (DEQ) with ReLU activation. They proved that the gradient descent converges to a globally optimal solution at a linear convergence rate for the quadratic loss function. This paper shows that this fact still holds for DEQs with any general activation that has bounded first and second derivatives. Since the new activation function is generally non-homogeneous, bounding the least eigenvalue of the Gram matrix of the equilibrium point is particularly challenging. To accomplish this task, we need to create a novel population Gram matrix and develop a new form of dual activation with Hermite polynomial expansion.

## 1  Introduction

Deep learning is a class of machine learning algorithms that uses multiple layers to progressively extract higher-level features from the raw input. For example, in image processing, lower layers may identify edges, while higher layers may identify the concepts relevant to a human such as digits or letters or faces. Deep neural networks have underpinned state of the art empirical results in numerous applied machine learning tasks (Krizhevsky et al., 2012). Understanding neural network learning, particularly its recent successes, commonly decomposes into the two main themes: (i) studying generalization capacity of the deep neural networks and (ii) understanding why efficient algorithms, such as stochastic gradient, find good weights. Though still far from being complete, previous work provides some understanding on generalization capability of deep neural networks. However, question (ii) is rather poorly understood. While learning algorithms succeed in practice, theoretical analysis is overly pessimistic. Direct interpretation of theoretical results suggests that when going slightly deeper beyond single layer networks, e.g. to depth-two networks with very few hidden units, it is hard to predict even marginally better than random (Daniely et al., 2013; Kearns & Valiant, 1994).

The standard approach to develop generalization bounds on deep learning (and machine learning) was developed in seminal papers by (Vapnik, 1998), and it is based on bounding the difference between the generalization error and the training error. These bounds are expressed in terms of the so called VC-dimension of the class. However, these bounds are very loose when the VC-dimension of the class can be very large, or even infinite. In 1998, several authors (Bartlett & Shawe-Taylor, 1999; Bartlett et al., 1998) suggested another class of upper bounds on generalization error that are expressed in terms of the empirical distribution of the margin of the predictor (the classifier). Later, Koltchinskii and Panchenko proposed new probabilistic upper bounds on generalization error of the combination of many complex classifiers such as deep neural networks (Koltchinskii & Panchenko, 2002). These bounds were developed based on the general results of the theory of Gaussian, Rademacher, and empirical processes in terms of general functions of the margins, satisfying a Lipschitz condition. They improved previously known bounds on the generalization error of convex combinations of classifiers. (Truong, 2022a) and Truong (2022b) have recently provided generalization bounds for learning with Markov dataset based on Rademacher and Gaussian complexity functions. The development of new symmetrization inequalities and contraction lemmas in high-dimensional

probability for Markov chains is a key element in these works. Several recent works have focused on gradient descent based PAC-Bayesian algorithms, aiming to minimise a generalization bound for stochastic classifiers (Biggs & Guedj, 2021; Dziugaite & Roy., 2017). Most of these studies use a surrogate loss to avoid dealing with the zero-gradient of the misclassification loss. There were some other works which use information-theoretic approaches to find PAC-bounds on generalization errors for machine learning (Esposito et al., 2021; Xu & Raginsky, 2017) and deep learning (Jakubovitz et al., 2018).

Recently, Deep Equilibrium Models (DEQs) (Bai et al., 2019) were introduced as a novel approach for modeling sequential data. In traditional deep sequence models, hidden layers often converge to fixed points, while DEQs directly find these equilibrium points through root-finding of implicit equations. Essentially, DEQs function as infinite-depth, weight-tied models with input injection. This framework has gained attention in a variety of applications, including computer vision (Bai et al., 2020; Xie et al., 2022), natural language processing (Bai et al., 2019), and inverse problems (Gilton et al., 2021), where DEQs have demonstrated competitive performance compared to state-of-the-art deep networks, often with significantly lower memory requirements. Despite the empirical success of DEQs, their theoretical understanding remains limited. The effectiveness of over-parameterization in optimizing feedforward neural networks has been demonstrated in several studies (Arora et al., 2019; Du et al., 2018; Li & Liang, 2018). A recent work (Nguyen, 2021) showed that gradient descent (GD) can converge to a global optimum when the width of the last hidden layer exceeds the number of training samples. This approach investigates properties at initialization and bounds the distance that GD travels from this initial point.

However, it remains unclear whether these results directly extend to DEQs. The implicit weight-sharing in DEQs introduces a dependence between initial random weights and features, making standard concentration methods ineffective in this context. Recently, Ling et al. (2022) examined the training dynamics of over-parameterized DEQs with ReLU activation. They introduced a novel probabilistic framework to address the challenges posed by weight-sharing and infinite depth. By imposing a condition on the initial equilibrium point, they showed that gradient descent converges to a globally optimal solution with linear convergence for the quadratic loss function. To achieve this, they developed a lower bound on the smallest eigenvalue of the Gram matrix for DEQs with ReLU activation. An interesting open question, however, is whether gradient descent will still converge at a linear rate for DEQs with activation functions other than ReLU.

It is crucial to explore alternative activation functions in DEQs because ReLU can lead to issues like dead neurons, gradient saturation, and instability in fixed-point iterations. Functions such as GELU, Swish, and tanh provide smoother gradients, better gradient flow, and more expressive non-linearities, which contribute to improved stability, faster convergence, and enhanced generalization. These alternatives address the limitations of ReLU, potentially improving model performance and making DEQs more effective at capturing complex patterns, especially in tasks that demand stable and efficient training.

In this paper, we demonstrate that the results obtained for DEQs with ReLU also hold for DEQs with general activation functions, provided they have bounded first and second derivatives. Many widely-used activation functions, such as $\max(x,0), 1/(1 + e^{-x}), \mathrm{erf}(x), x/\sqrt{1 + x^2}, \sin(x), \tanh(x)$, satisfy these boundedness conditions. Since these new activation functions do not generally possess the homogeneous property of ReLU, we propose a novel population Gram matrix for DEQs with general activations and introduce a new form of dual activation, incorporating Hermite polynomial expansion, to address these challenges.

## 2 Problem setting

We consider the same model as Ling et al. (2022). However, different from Ling et al. (2022), we assume that the activation function, $\varphi$, satisfies some constraints in the first and second derivatives. These properties can be observed in many common activation functions. More specifically, we define a vanilla deep equilibrium model (DEQ) with the $l$-th layer transforming as

$$\mathbf{T}^{(l)} = \varphi(\mathbf{W}\mathbf{T}^{(l-1)} + \mathbf{U}\mathbf{X}) \tag{1}$$

where $\mathbf{X} = [\mathbf{x}_1, \mathbf{x}_2, \cdots, \mathbf{x}_n] \in \mathbb{R}^{d \times n}$ denotes the training inputs, $\mathbf{U} \in \mathbb{R}^{m \times d}$ and $\mathbf{W} \in \mathbb{R}^{m \times m}$ are trainable weight matrices, and $\mathbf{T}^{(l)} \in \mathbb{R}^{m \times n}$ is the output feature at the $l$-th hidden layer. If we were to repeat this

update an infinite number of times, we would essentially be modeling an infinitely deep network of the form above. In practice, what we find is that for most "typical" deep layers the valued actually converge to a fixed point or equilibrium point (Bai et al., 2019). In Theorem 3 below, we show that $\{\mathbf{T}^{(l)}\}_{l=1}^{\infty}$ converges under the condition $\|\mathbf{W}\|_2 < 1/L$ and and the activation function $\varphi$ is $L$-bounded for some $L \in \mathbb{R}_+$. The output of the last hidden layer is defined by $\mathbf{T}^* := \lim_{l \to \infty} \mathbf{T}^{(l)}$. Therefore, instead of running infinitely deep layer-by-layer forward propagation, $\mathbf{T}^*$ can be calculated by directly solving the equilibrium point of the following equation

$$\mathbf{T}^* = \varphi(\mathbf{W}\mathbf{T}^* + \mathbf{U}\mathbf{X}). \tag{2}$$

Let $\mathbf{y} = [y_1, y_2, \cdots, y_n] \in \mathbb{R}^n$ denote the labels, and $\hat{\mathbf{y}}(\boldsymbol{\theta}) = \mathbf{a}^T\mathbf{T}^*$ be the prediction function with $\mathbf{a} \in \mathbb{R}^m$ being a trainable vector and $\theta = \text{vec}(\mathbf{W}, \mathbf{U}, \mathbf{a})$. Our target is to minimize the empirical risk with the quadratic loss function:

$$\Phi(\boldsymbol{\theta}) = \frac{1}{2}\|\hat{\mathbf{y}}(\boldsymbol{\theta}) - \mathbf{y}\|_2^2. \tag{3}$$

To optimize this loss function, we use the gradient descent update $\boldsymbol{\theta}(\tau + 1) = \boldsymbol{\theta}(\tau) - \eta\nabla\Phi(\boldsymbol{\theta}(\tau))$, where $\eta$ is the learning rate and $\boldsymbol{\theta}(\tau) = \text{vec}(\mathbf{W}(\tau), \mathbf{U}(\tau), \mathbf{a}(\tau))$. For notational simplicity, we omit the superscript and denote $\mathbf{T}$ to be the equilibrium $\mathbf{T}^*$ when it is clear from the context. Moreover, the Gram matrix of the equilibrium point is defined by $\mathbf{G}(\tau) := \mathbf{T}^T(\tau)\mathbf{T}(\tau)$ and we define its least eigenvalue by $\lambda_\tau = \lambda_{\min}(\mathbf{G}(\tau))$. In this paper, for brevity we denote by $\mathbf{G} = \mathbf{G}(0)$.

**Definition 1.** *An activation $\varphi : \mathbb{R} \to \mathbb{R}$ is $L$-bounded if it is twice continuously differentiable and* $\max\{|\varphi(0)|, \|\varphi'\|_\infty, \|\varphi''\|_\infty\} \leq L$.

In this paper, we assume that $\varphi(\cdot)$ is $L$-bounded. Many popular activation functions such as $\max(x, 0), 1/(1 + e^{-x}), \text{erf}(x), x/\sqrt{1 + x^2}, \sin(x), \tanh(x)$ satisfy the boundedness requirements.

**Definition 2.** *Two vectors $\mathbf{a}, \mathbf{b} \in \mathbb{R}^n$ are said to be parallel, denoted $\mathbf{a} \parallel \mathbf{b}$, if there exists a scalar $\kappa \in \mathbb{R}$ such that $\mathbf{a} = \kappa\mathbf{b}$. If $\mathbf{a}$ and $\mathbf{b}$ are not parallel, we write $\mathbf{a} \nparallel \mathbf{b}$.*

Additionally, we adopt similar assumptions regarding the random initialization and input data as those used in Ling et al. (2022):

- **Assumption 1** (Random initialization). Assume that $\sigma_w^2 < \frac{1}{8L^2}$. In addition, $\mathbf{W}$ is initialized with an $m \times m$ matrix with i.i.d. entries $\mathbf{W}_{ij} \sim \mathcal{N}(0, 2\sigma_w^2/m)$, $\mathbf{U}$ is initialized with an $m \times d$ matrix with i.i.d. entries $\mathbf{U}_{ij} \sim \mathcal{N}(0, 2/m)$, and $\mathbf{a}$ is initialized with a random vector with i.i.d. entries $\sim \mathcal{N}(0, 1/m)$.

- **Assumption 2** (Input data). We assume that (i) $\|\mathbf{x}_i\|_2 = \sqrt{d}$ for all $i \in [n]$ and $\mathbf{x}_i \nparallel \mathbf{x}_j$ for all $i \neq j$; (ii) the labels satisfy $|y_i| = O(1)$ for all $i \in [n]$.

## 3 Main results

In this paper, we show that if the learning rate is small enough, the loss converges to a global minimum at linear rate. The result is as follows.

**Theorem 3.** *Consider a DEQ. Let $\delta$ be a constant such that $\|\mathbf{W}(0)\|_2 + \delta < 1/L$. Denote by $\bar{\rho}_w = \|\mathbf{W}(0)\|_2 + \delta, \bar{\rho}_u = \|\mathbf{U}(0)\|_2 + \delta, \bar{\rho}_a = \|\mathbf{a}(0)\|_2 + \delta$ and define*

$$c_a = \frac{L\bar{\rho}_u}{1 - L\bar{\rho}_w}, \qquad c_u = \frac{L\bar{\rho}_a}{1 - L\bar{\rho}_w}, \qquad c_m = \frac{|\varphi(0)|\sqrt{mn}}{1 - L\bar{\rho}_w}. \tag{4}$$

*In addition, assume at initialization that*

$$\lambda_0 \geq \frac{4}{\delta} \max \left\{ c_u \big( c_a \|\mathbf{X}\|_F + c_m \big), c_u \|\mathbf{X}\|_F, c_a \|\mathbf{X}\|_F + c_m \right\} \|\hat{\mathbf{y}}(0) - \mathbf{y}\|, \tag{5}$$

$$\lambda_0^{3/2} \geq \frac{4(2 + \sqrt{2})L}{(1 - L\bar{\rho}_w)} \left[ c_u \big( c_a \|\mathbf{X}\|_F + c_m \big)^2 + c_u \|\mathbf{X}\|_F^2 \right] \|\hat{\mathbf{y}}(0) - \mathbf{y}\|_2, \tag{6}$$

$$\lambda_0 \geq 8 \left[ c_u^2 \big( c_a \|\mathbf{X}\|_F + c_m \big)^2 + c_u^2 \|\mathbf{X}\|_F^2 \right] \tag{7}$$

*where $\lambda_0$ is the least eigenvalue of $\mathbf{G}(0) = \mathbf{T}(0)^T \mathbf{T}(0)$. Then, if the learning rate satisfies*

$$\eta < \min \left( \frac{2}{\lambda_0}, \frac{2[c_u^2(c_a\|\mathbf{X}\|_F + c_m)^2 + c_u^2\|\mathbf{X}\|_F^2]}{c_u^2(c_a\|\mathbf{X}\|_F + c_m)^2 + c_u^2\|\mathbf{X}\|_F^2 + (c_a\|\mathbf{X}\|_F + c_m)^2} \right),$$

*for every $\tau \geq 0$, the following hold:*

- $\|\mathbf{W}(\tau)\|_2 < 1/L$, *i.e., the equilibrium points always exists,*

- $\lambda_\tau \geq \frac{1}{2}\lambda_0$, *and*

$$\|\nabla_\theta \Phi(\theta(\tau))\|_2^2 \geq \lambda_0 \Phi(\theta(\tau)). \tag{8}$$

- *The loss converges to a global minimum, i.e., $\hat{\mathbf{y}}(\theta(\tau)) \to \mathbf{y}$, as*

$$\Phi(\theta(\tau)) \leq \left( 1 - \eta \frac{\lambda_0}{2} \right)^\tau \Phi(\theta(0)). \tag{9}$$

The existence of $\delta$ such that $\|\mathbf{W}(0)\|_2 + \delta < 1/L$ follows from Assumption 1, which states that $\sigma_w^2 < \frac{1}{8L^2}$, and is supported by the following lemma:

**Lemma 4.** *(Ling et al., 2022, Lemma 1) Let $\mathbf{W}$ be an $m \times m$ random matrix with i.i.d. entries $\mathbf{W}_{ij} \sim \mathcal{N}(0, 2\sigma_w^2/m)$. With probability at least $1 - \exp(-\Omega(m))$, it holds that $\|\mathbf{W}\|_2 \leq 2\sqrt{2}\sigma_w$.*

The main challenge now is to find an appropriate initialization such that $\lambda_0$ satisfies all the conditions in Theorem 3. Estimating $\lambda_0$ directly is difficult, and a common strategy is to establish a concentration inequality between *the initial empirical Gram matrix* $\mathbf{G}$ and a new matrix with a more easily estimable least eigenvalue. This new matrix is referred to as *the population Gram matrix* and is denoted by $\mathbf{K}$ Ling et al. (2022). However, due to the non-homogeneity of the new activation function $\varphi$, bounding $\lambda_0$ becomes more challenging than in the case of ReLU networks, as discussed in Ling et al. (2022). The non-homogeneity of the activation functions makes the design techniques for $\mathbf{K}$ presented in (Ling et al., 2022, Definition 1) inapplicable. For example, (Ling et al., 2022, Eq. 11) is only valid for the ReLU function.

In Section 4, we propose a new method to create the population Gram matrix $\mathbf{K}$ for DEQs with general Lipschitz activation functions. By using our new form of dual activation and Hermite polynomial expansion, we can prove that $\mathbf{K}$ is symmetric positive definite. In addition, we show that with probability at least $1 - t$, $\lambda_0 \geq \frac{m}{2}\lambda_*$ provided that $m = \Omega\big(\frac{n^2}{\lambda_*^2} \log \frac{n}{t}\big)$ where $\lambda_*$ is the least eigenvalue of $\mathbf{K}$ (cf. Section 7). Thanks to this, we can show that there exists a weight initialization algorithm (WIAL) such that all the conditions of Theorem 3 hold for over-parameterized DEQs, specifically when $m = \Omega\big(\frac{n^3}{\lambda_*^2} \log \frac{n}{t}\big)$ (cf. Section 7). Hence, by (9) in Theorem 3, the gradient descent algorithm converges to a global optimum at a linear rate for the over-parametrized DEQs if the number of repetitions in (1) sufficiently large. This intriguing result is further confirmed by our numerical experiments on real datasets, including MNIST and CIFAR-10, presented in Section 8.

# 4 A novel design of the population Gram matrix K

The key approach in lower bounding $\lambda_0$ is to design a population Gram matrix $\mathbf{K}$ in such a way that we can lower bound $\lambda_0$ by the least eigenvalue of $\mathbf{K}$ and that $\mathbf{K}$ is symmetric positive definite. This novel population Gram matrix is developed through our introduction of a new form of dual activation.

First, we define a new class of dual activation functions $\tilde{Q}_{\alpha,\beta} : [-1,1] \to \mathbb{R}$ for all pairs $(\alpha,\beta) \in \mathbb{R}_+^2$.

**Definition 5.** *Let*

$$q := \sqrt{\frac{2}{\sqrt{2\pi}} \int_{-\infty}^{\infty} \varphi^2(z) \exp\left(-\frac{z^2}{2}\right) dz}.$$

*For each pair $(\alpha,\beta)$, define*

$$\tilde{Q}_{\alpha,\beta}(x) := \frac{1}{\alpha\beta q^2} \mathbb{E}_{(a,b)^T \sim \mathcal{N}\left(0, \begin{bmatrix} 1 & x \\ x & 1 \end{bmatrix}\right)} \left[\varphi(\alpha a)\varphi(\beta b)\right], \quad \forall |x| \leq 1. \tag{10}$$

If $\varphi(x) = \max\{x,0\}$ (ReLU), then $\tilde{Q}_{\alpha,\beta}(x) = \bar{Q}(x)$ for all $(\alpha,\beta) \in \mathbb{R}_+^2$, where

$$\bar{Q}(x) := \mathbb{E}_{(a,b)^T \sim \mathcal{N}\left(0, \begin{bmatrix} 1 & x \\ x & 1 \end{bmatrix}\right)} \left[\varphi(a)\varphi(b)\right]$$

is the dual activation defined in (Daniely et al., 2016, Sec. 3.2).

Now, we provide a novel design of the population Gram matrix $\mathbf{K}$ based on this new dual activation function.

**Definition 6.** *Given the training input $\mathbf{X} := [\mathbf{x}_1, \mathbf{x}_2, \cdots, \mathbf{x}_n]$ that satisfies Assumption 2, let*

$$Q_{ij}(x) := \tilde{Q}_{\sqrt{2\left(\frac{\sigma_w^2}{m}\mathbb{E}[\mathbf{G}_{ii}]+1\right)}, \sqrt{2\left(\frac{\sigma_w^2}{m}\mathbb{E}[\mathbf{G}_{jj}]+1\right)}}(x), \qquad \forall x \in \mathbb{R}. \tag{11}$$

*We define the population Gram matrices $\mathbf{K}^{(l)}$ of each layer recursively as*

$$\rho_{ij}^{(0)} = 0, \tag{12}$$

$$\rho_{ii}^{(l)} = 2q^2 \sigma_w^2 \rho_{ii}^{(l-1)} Q_{ii}(1) + 1, \tag{13}$$

$$\rho_{ij}^{(l)} = \sqrt{\rho_{ii}^{(l)} \rho_{jj}^{(l)}}, \quad i \neq j \tag{14}$$

$$\mathbf{K}^{(0)} = 0, \tag{15}$$

$$\nu_{ij}^{(l)} = \frac{\sigma_w^2 \mathbf{K}_{ij}^{(l-1)} + d^{-1}\mathbf{x}_i^T \mathbf{x}_j}{\sqrt{\left(\sigma_w^2 \mathbf{K}_{ii}^{(l-1)} + 1\right)\left(\sigma_w^2 \mathbf{K}_{jj}^{(l-1)} + 1\right)}} \tag{16}$$

$$\mathbf{K}_{ij}^{(l)} = 2q^2 \rho_{ij}^{(l)} Q_{ij}(\nu_{ij}^{(l)}) \tag{17}$$

*for all $l \geq 1$ and $i,j \in [n] \times [n]$.*

The next result shows that $\lambda_0$ can be lower-bounded via the least eigenvalue of the population matrix $\mathbf{K}$, where $\mathbf{K} = \lim_{l \to \infty} \mathbf{K}^{(l)}$. The existence of this limit will be proved in Proposition 13.

**Theorem 7.** *If $m = \Omega\left(\frac{n^2}{\lambda_*^2} \log \frac{n}{t}\right)$, with probability at least $1 - t$, it holds that*

$$\lambda_0 \geq \frac{m}{2}\lambda_*. \tag{18}$$

Based on Theorem 7, we can show that there exists a weight initialization algorithm (WIAL) such that all the conditions in (5)–(7) of Theorem 3 are satisfied for sufficiently large $m$, provided that $\lambda_* > 0$ (cf. Section 7). The following result establishes a sufficient condition for $\lambda_* > 0$, or equivalently, for $\mathbf{K}$ to be strictly positive definite.

**Theorem 8.** *Assume that there exists a polynomial expansion of $\tilde{Q}_{\alpha,\alpha}$ satisfying:*

$$\tilde{Q}_{\alpha,\alpha}(x) = \sum_{r=0}^{\infty} \mu_{r,\alpha}^2(\varphi) x^r \tag{19}$$

*for all $\alpha > 0$ such that $\sup\{r : \min_{\alpha \in \left[2, 2\left(\sigma_w^2 \frac{8L^2 d + 2L^2}{1 - 8L^2\sigma_w^2} + 1\right)\right]} \mu_{r,\alpha}^2(\varphi) > 0\} = \infty$. Then, $\mathbf{K}$ is strictly positive definite with the least eigenvalue satisfying $\lambda_* \geq \lambda_0^*$ for some $\lambda_0^* > 0$ which does not depend on $m$.*

## 5 Proof of Theorem 7

To prove Theorem 7, we first present some auxiliary results based on the population Gram matrix $\mathbf{K}$ from Definition 6. The proofs of these lemmas and propositions can be found in the Appendix.

**Lemma 9.** *Recall the definition of $\tilde{Q}_{\alpha,\beta}$ in Definition 5. Then, the following hold for all $\alpha > 0, \beta > 0$:*

$$\left|\tilde{Q}_{\alpha,\beta}(x)\right| \leq \sqrt{\tilde{Q}_{\alpha,\alpha}(1)\tilde{Q}_{\beta,\beta}(1)}, \tag{20}$$

$$\left|\tilde{Q}_{\alpha,\beta}(x)\right| \leq \frac{4L^2}{q^2}, \qquad \forall |x| \leq 1. \tag{21}$$

*In addition, $\tilde{Q}_{\alpha,\beta}(\cdot)$ is $\frac{2L^2 \max\{\alpha+1, \beta+1\}^2}{q^2}$-Lipchitz for any fixed positive pair $(\alpha, \beta)$.*

**Lemma 10.** *(Ling et al., 2022, Proof of Lemma 4) For $l \geq 1$, $\mathbf{G}_{ij}^{(l+1)}$ can be reconstructed as $\mathbf{G}_{ij}^{(l+1)} = \varphi(\mathbf{Mh}_{l+1})^T \varphi(\mathbf{Mh}'_{l+1})$ such that*

- *(i) $\mathbf{h}_{l+1}^T \mathbf{h}'_{l+1} = \frac{\sigma_w^2}{m} \mathbf{G}_{ij}^{(l)} + \frac{1}{d}\mathbf{x}_i^T \mathbf{x}_j$,*

- *(ii) $\mathbf{M} \in \mathbb{R}^{m \times (2l+d+2)}$ is a rectangle matrix, and the entries of $\mathbf{M}$ are i.i.d. from $\mathcal{N}(0,2)$ conditioning on previous layers.*

**Lemma 11.** *For the given setting, we have*

$$\rho_{ii}^{(l)} = \sigma_w^2 \mathbf{K}_{ii}^{(l-1)} + 1, \tag{22}$$

$$\rho_{ij}^{(l)} \nu_{ij}^{(l)} = \sigma_w^2 \mathbf{K}_{ij}^{(l-1)} + d^{-1}\mathbf{x}_i^T \mathbf{x}_j, \qquad \forall i, j, \tag{23}$$

*and*

$$\nu_{ij}^{(l)} = \begin{cases} \frac{Q_{ij}\left(\nu_{ij}^{(l-1)}\right)/\sqrt{Q_{ii}(1)Q_{jj}(1)}\sqrt{(\rho_{ii}^{(l)}-1)(\rho_{jj}^{(l)}-1)}+d^{-1}\mathbf{x}_i^T\mathbf{x}_j}{\sqrt{\rho_{ii}^{(l)}\rho_{jj}^{(l)}}}, & i \neq j \\ 1, & i = j \end{cases}. \tag{24}$$

*In addition, we also have*

$$\left|\nu_{ij}^{(l)}\right| \leq 1 \tag{25}$$

*for all $i, j \in [n] \times [n]$ and $l \geq 0$.*

Building on Lemma 9, Lemma 10, and Lemma 11, we can prove the following three propositions.

**Proposition 12.** *Under Assumptions 1 and 2 with probability at least $1 - n^2 \exp(-\Omega(m))$, it holds that*

$$\frac{1}{m}\left\|\mathbf{G} - \mathbf{G}^{(l)}\right\|_F = O\left(n\left(2L\sqrt{2}\sigma_w\right)^l\right). \tag{26}$$

**Proposition 13.** *Under Assumptions 1 and 2, we have the following relationship:*

$$\left\| \mathbf{K} - \mathbf{K}^{(l)} \right\|_F = O\big(n(8L^2\sigma_w^2)^l\big),$$

*which implies that as $l \to \infty$, $\mathbf{K}^{(l)} \to \mathbf{K}$, where $\mathbf{K} \in \mathbb{R}^{n \times n}$ is a matrix with entries*

$$\mathbf{K}_{ij} = 2q^2 Q_{ij}(\nu_{ij})\sqrt{\rho_{ii}\rho_{jj}} \tag{27}$$

*where*

$$\nu_{ij} = \begin{cases} \dfrac{Q_{ij}(\nu_{ij})/\sqrt{Q_{ii}(1)Q_{jj}(1)}\sqrt{(\rho_{ii}-1)(\rho_{jj}-1)}+d^{-1}\mathbf{x}_i^T\mathbf{x}_j}{\sqrt{\rho_{ii}\rho_{jj}}}, & i \neq j \\ 1, & i = j \end{cases}. \tag{28}$$

*Here,*

$$\rho_{ii} = \frac{1}{1 - 2q^2\sigma_w^2 Q_{ii}(1)}. \tag{29}$$

**Proposition 14.** *Under Assumptions 1 and 2, with probability at least $1 - n^2 l \exp\big\{ -\Omega(8^l L^{2l}\sigma_w^{2l}mnL^2) + O(l^2)\big\}$, it holds that*

$$\left\| \frac{1}{m}\mathbf{G}^{(l)} - \mathbf{K}^{(l)} \right\|_F = O\left(n(2L\sqrt{2}\sigma_w)^l\right). \tag{30}$$

Finally, by combining Propositions 13–14, we can bound $\lambda_0$ in terms of the smallest eigenvalue of the population matrix $\mathbf{K}$ as follows.

*Proof of Theorem 7.* From Propositions 13–14, with probability at least $1 - n^2 \exp\big(-\Omega(m8^l L^{2l}\sigma_w^{2l}) + O(l^2)\big)$, it holds that

$$\begin{aligned}
\left\| \frac{1}{m}\mathbf{G} - \mathbf{K} \right\|_F &\leq \frac{1}{m}\left\| \mathbf{G} - \mathbf{G}^{(l)} \right\|_F + \left\| \frac{1}{m}\mathbf{G}^{(l)} - \mathbf{K}^{(l)} \right\| + \left\| \mathbf{K} - \mathbf{K}^{(l)} \right\|_F \\
&= O\left(n\left(2L\sqrt{2}\sigma_w\right)^l\right) + O\left(n\left(2L\sqrt{2}\sigma_w\right)^l\right) + O\left(n(8L^2\sigma_w^2)^l\right) \\
&= O\left(n\left(2L\sqrt{2}\sigma_w\right)^l\right),
\end{aligned} \tag{31}$$

where (31) follows from $\sigma_w^2 < 1/(8L^2)$.

Next, we fix $l$ to omit the explicit dependence on $l$. Specifically, let

$$l = \Theta(\log(2\lambda_*^{-1}n)/\log(\sqrt{2}/(4L\sigma_w)),$$

then from (31), we have

$$\left\| \frac{1}{m}\mathbf{G} - \mathbf{K} \right\|_F \leq \frac{\lambda_*}{2}.$$

It is easy to prove by induction that $\mathbf{K}$ is symmetric. Therefore, by Weyl's inequality (Ling et al., 2022, Lemma 5), it holds that

$$\max_{i \in [r]} \left| \lambda_i\left(\frac{1}{m}\mathbf{G}\right) - \lambda_i(\mathbf{K}) \right| \leq \left\| \frac{1}{m}\mathbf{G} - \mathbf{K} \right\|_2 \leq \left\| \frac{1}{m}\mathbf{G} - \mathbf{K} \right\|_F \leq \frac{\lambda_*}{2}.$$

Now, by choosing $i_0 := \arg\min_i \lambda_i(\mathbf{K})$, we have

$$\lambda_{i_0}(\mathbf{K}) = \lambda_* \tag{32}$$

and

$$\left| \frac{1}{m} \lambda_{\min}(\mathbf{G}) - \lambda_* \right| \leq \frac{\lambda_*}{2}. \tag{33}$$

It follows from (32) and (33) that

$$\lambda_0 = \lambda_{\min}(\mathbf{G}) \geq \frac{m}{2} \lambda_*.$$

Consequently, w.p. $\geq 1 - t$, we have $\lambda_0 \geq \frac{m}{2} \lambda_*$ provided that $m = \Omega\left(\frac{n^2}{\lambda_*^2} \log \frac{n}{t}\right)$. $\qquad\square$

## 6 Checking the conditions of Theorem 8

In this section, we will demonstrate how the conditions in Theorem 8 hold for several common activation functions. First, we recall the definition of a traditional dual activation function, denoted $\hat{\varphi}$, associated with $\varphi$ in (Daniely et al., 2016, Sect. 4.2):

$$\hat{\varphi}(x) = \mathbb{E}_{(u,v)\sim\mathcal{N}\left(0, \begin{bmatrix} 1 & x \\ x & 1 \end{bmatrix}\right)} [\varphi(u)\varphi(v)].$$

By following a similar proof as in (Daniely et al., 2016, Lemma 11), it can be shown that the new activation function (see Definition 5) satisfies

$$\tilde{Q}_{\alpha,\alpha}(x) = \frac{1}{q^2\alpha^2} \sum_{n=1}^{\infty} a_n^2 \alpha^{2n} x^n \tag{34}$$

if either $\varphi(x) = \sum_{n=1}^{\infty} a_n h_n(x)$ (Hermite polynomial expansion) or $\hat{\varphi}(x) = \sum_{n=1}^{\infty} a_n^2 x^n$. Here, $h_0, h_1, \cdots$ are the normalized Hermite polynomials, which are obtained by applying the Gram-Schmidt process to the sequence $1, x, x^2, \cdots$ with respect to the inner product $\langle f, g \rangle = \frac{1}{\sqrt{2\pi}} \int_{-\infty}^{\infty} f(x)g(x)e^{-x^2/2}dx$ (cf. Daniely et al. (2016)).

In the following, we apply (34) and show how the conditions in Theorem 8 are satisfied.

**Example 15.** *Consider the sine activation, $\varphi(x) = \sin(ax)$. By (Daniely et al., 2016, Sect. 8), we have*

$$\hat{\varphi}(x) = e^{-a^2} \sinh(a^2 x).$$

*By Taylor's expansion of* sinh *function, i.e.,*

$$\sinh(x) = \sum_{r=0}^{\infty} \frac{1}{(2r+1)!} x^{2r+1}.$$

*Hence, from* (34) *we have*

$$\tilde{Q}_{\alpha,\alpha}(x) = \frac{1}{q^2\alpha^2} e^{-a^2} \sum_{r=0}^{\infty} \frac{a^{4r+2}\alpha^{4r+2}}{(2r+1)!} x^{2r+1},$$

*which leads to*

$$\mu_{r,\alpha}^2(\varphi) = \begin{cases} \frac{1}{q^2\alpha^2} e^{-a^2} \frac{a^{2r}\alpha^{2r}}{r!} & r \mod 2 = 1 \\ 0 & otherwise \end{cases}.$$

*This means that the condition in Theorem 8 is satisfied.*

**Example 16.** *Consider the tanh activation function, $\varphi(x) = \frac{e^x - e^{-x}}{e^x + e^{-x}}$. By (Szego, 1959, Eq. 8.23.4), $\varphi(x)$ can be uniquely described in the basis of Hermite polynomials,*

$$\varphi(x) = \sum_{n=1}^{\infty} a_n h_n(x)$$

*where*

$$|a_n| = \frac{1}{\sqrt{\pi} 2^n n!} \frac{\Gamma\left(\frac{n}{2} + 1\right)}{\Gamma(n+1)} \exp\left(-\frac{\pi\sqrt{2n}}{2}\right).$$

*Hence, from (34), we obtain*

$$\tilde{Q}_{\alpha,\alpha}(x) = \frac{1}{q^2 \alpha^2} \sum_{n=1}^{\infty} a_n^2 \alpha^{2n} x^n,$$

*so we have*

$$\mu_{r,\alpha}^2(\varphi) = \frac{1}{q^2 \alpha^2} a_n^2 \alpha^{2n}$$

*This means that the condition in Theorem 8 is satisfied.*

**Example 17.** *Consider the sigmoid activation function $\varphi(x) = \frac{1}{1+e^{-x}}$. It is known that*

$$\varphi(x) = \frac{1 + \tanh(x/2)}{2}.$$

*Hence, by using similar arguments as Example 16, we can prove that the condition in Theorem 8 is also satisfied.*

# 7 Weight Initialisation Algorithm

Before proposing an algorithm to initialise weights, we introduce some initial results.

**Lemma 18.** *(Vershynin, 2018, Theorem 4.4.5) For a random matrix $\mathbf{A} \in \mathbb{R}^{n \times m}$ with $\mathbf{A}_{ij} \sim \mathcal{N}(0,1)$, it holds that*

$$\|\mathbf{A}\|_2 \leq C(\sqrt{m} + \sqrt{n} + t) \tag{35}$$

*with probability $1 - 2e^{-t^2}$, where $C$ is some constant.*

**Lemma 19.** *For any fixed $t \in \mathbb{R}_+$, it holds that*

$$\|\hat{\mathbf{y}}(0) - \mathbf{y}\| = O(\sqrt{n}) \tag{36}$$

*with probability at least $1 - t$.*

A weight initialisation algorithm (WIALG) is as follows.

- **Initialise:** $m = n, \sigma_w^2 = \frac{1}{96L^2}$.

- **Step 1:**

    - Generate a matrix $\mathbf{W} \in \mathbb{R}^{m \times m}$ where $\mathbf{W}_{ij} \sim \mathcal{N}\left(0, \frac{2\sigma_w^2}{m}\right)$.
    - Generate a matrix $\mathbf{U} \in \mathbb{R}^{m \times d}$ where $\mathbf{U}_{ij} \sim \mathcal{N}\left(0, \frac{2}{m}\right)$.
    - Generate a vector $\mathbf{a} \in \mathbb{R}^m$ where $\mathbf{a}_i \sim \mathcal{N}\left(0, \frac{1}{m}\right)$.

- **Step 2:**

- Find a fixed-point $\mathbf{T}$ of the equation $\mathbf{T} = \varphi(\mathbf{WT} + \mathbf{UX})$ by using Anderson acceleration method Walker & Ni (2011).

- Estimate $\frac{\mathbb{E}[\mathbf{G}_{ii}]}{m}$ by using the Monte-Carlo method. Note that by our Assumption 2, $\mathbb{E}[\mathbf{G}_{ii}]$ does not depend on $i$, so we only need to estimate $\frac{\mathbb{E}[\mathbf{G}_{11}]}{m}$.

- Set $\hat{\mathbf{y}}(0) = \mathbf{a}^T \mathbf{T}$.

- **Step 3:**

  - Recursively construct a sequence $\mathbf{K}^{(l)}$ by using (12)–(17) until $\|\mathbf{K}^{(l)} - \mathbf{K}^{(l-1)}\|_F \leq \varepsilon$ for some small value $\varepsilon > 0$.

  - Estimate the least eigenvalue $\lambda_*$ of $\mathbf{K}^{(l)}$.

- **Step 4:** Check the following conditions:

$$\frac{m}{2}\lambda_* \geq \frac{4}{\delta} \max\left\{ c_u\big(c_a\|\mathbf{X}\|_F + c_m\big), c_u\|\mathbf{X}\|_F, c_a\|\mathbf{X}\|_F + c_m \right\} \|\hat{\mathbf{y}}(0) - \mathbf{y}\|, \tag{37}$$

$$\left(\frac{m}{2}\lambda_*\right)^{3/2} \geq \frac{4(2 + \sqrt{2})L}{(1 - L\bar{\rho}_w)}\left[ c_u\big(c_a\|\mathbf{X}\|_F + c_m\big)^2 + c_u\|\mathbf{X}\|_F^2 \right] \|\hat{\mathbf{y}}(0) - \mathbf{y}\|_2, \tag{38}$$

$$\frac{m}{2}\lambda_* \geq 8\left[ c_u^2\big(c_a\|\mathbf{X}\|_F + c_m\big)^2 + c_u^2\|\mathbf{X}\|_F^2 \right], \tag{39}$$

where $c_u, c_a, c_m, \bar{\rho}_w$ are defined in Theorem 3.

- **Step 5:** If all the conditions (37)–(39) hold, we STOP the initialisation. Otherwise, we increase $m = m + 10$ and REPEAT Step 1.

**Theorem 20.** *For DEQs with $\sigma(0) = 0$, WIALG will STOP with probability $1 - t$ at $m = \Omega\big(\frac{n^3}{(\lambda_*)^2} \log \frac{n}{t}\big)$.*

A detailed proof of Theorem 20 can be found in Appendix I. Finally, by combining Theorem 20 with Theorem 7, we conclude that if $m = \Omega\big(\frac{n^3}{(\lambda_0^*)^2} \log \frac{n}{t}\big)$ and $n$ sufficiently large, then with probability at least $1 - t$, all the conditions in (5)-(7) from Theorem 3 are satisfied.

## 8  Numerical Results

In this section, we conduct experiments to validate Theorem 3. Specifically, we evaluate the performance of the DEQ model on the MNIST and CIFAR-10 datasets. We adopt Gaussian initialization as stated in Assumption 1 and normalize each data point according to Assumption 2.

For the first experiment, we vary the parameter $m$ and plot the training dynamics for both MNIST and CIFAR-10 when the activation function $\varphi$ is the tanh function ($L = 1$). As shown in Fig. 1, when $m$ is sufficiently large and $\tau$ is sufficiently large, the curves approach straight lines. This observation further confirms that equation (9) holds true.

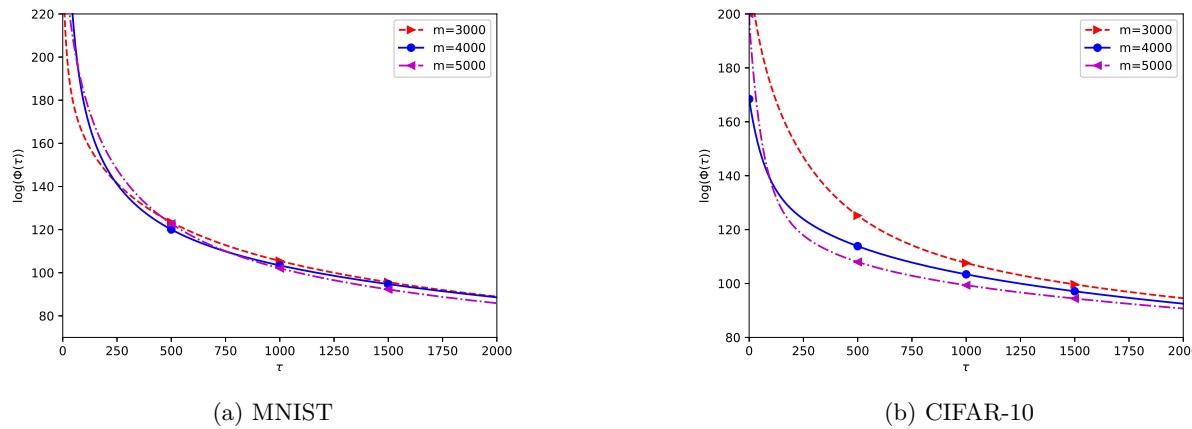

(a) MNIST

(b) CIFAR-10

Figure 1: Training dynamics at different values of $m$.

In the second experiment, we vary the activation function and plot the training dynamics for MNIST and CIFAR-10 with $m = 3000$ for several activation functions, including ReLU, Sin, Identity, Swish, and Mish. While the activation functions Swish and Mish do not have bounded first derivatives and are thus not $L$-bounded, we observe from Fig. 2 that when $m$ is sufficiently large and $\tau$ is sufficiently large, all the networks converge to the global optimum at a linear rate. This suggests that the $L$-bounded condition is a sufficient, but not necessary, condition for the linear convergence rate of the squared loss.

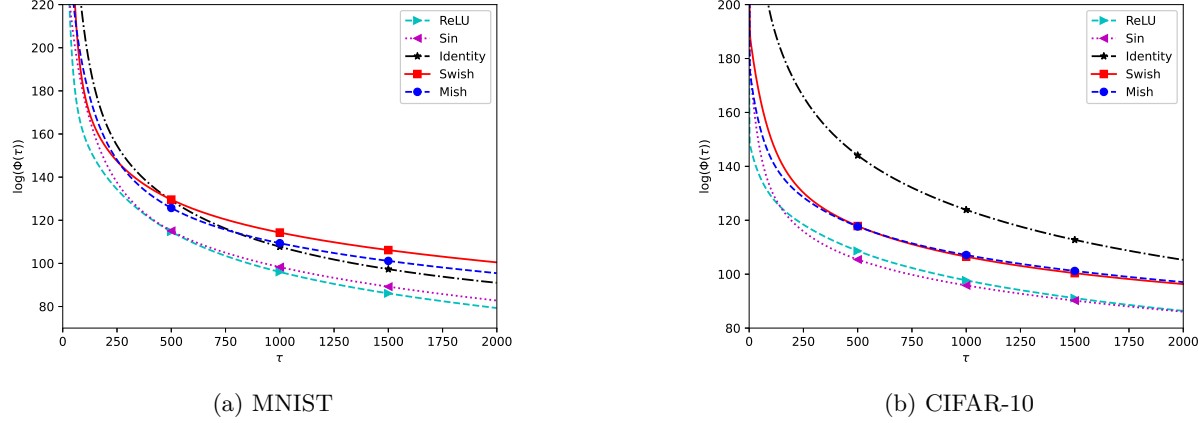

(a) MNIST

(b) CIFAR-10

Figure 2: Training dynamics for different activation functions.

## 9 Conclusion

In this paper, we have shown that gradient descent converges to a globally optimal solution at a linear convergence rate for the quadratic loss function in the context of over-parameterized DEQs with $L$-bounded activation functions. This compelling result is further supported by our numerical experiments on the MNIST and CIFAR-10 datasets. To overcome the technical challenges introduced by the non-homogeneity of activation functions, we propose a novel population Gram matrix and introduce a new form of dual activation based on Hermite polynomial expansion. An exciting direction for future research is to explore whether the linear convergence rate property holds for other classes of activation functions.

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

## A  Proof of Lemma 9

Observe that

$$
\begin{aligned}
\left|\tilde{Q}_{\alpha,\beta}(x)\right| &\leq \frac{1}{\alpha\beta q^2} \mathbb{E}_{(a,b)^T \sim \mathcal{N}\left(0, \begin{bmatrix} 1 & x \\ x & 1 \end{bmatrix}\right)} \left|\varphi(\alpha a)\varphi(\beta b)\right| \\
&= \frac{1}{\alpha\beta q^2} \mathbb{E}_{(u,v)^T \sim \mathcal{N}\left(0, \begin{bmatrix} \alpha^2 & x\alpha\beta \\ x\alpha\beta & \beta^2 \end{bmatrix}\right)} \left|\varphi(u)\varphi(v)\right| \\
&\leq \frac{1}{\alpha\beta} \sqrt{\frac{1}{q^2}\mathbb{E}_{a\sim\mathcal{N}(0,\alpha^2)}[\varphi^2(a)]} \sqrt{\frac{1}{q^2}\mathbb{E}_{b\sim\mathcal{N}(0,\beta^2)}[\varphi^2(b)]} \\
&= \sqrt{\tilde{Q}_{\alpha,\alpha}(1)\tilde{Q}_{\beta,\beta}(1)},
\end{aligned}
$$

(40)

(41)

where (40) follows from Cauchy–Schwarz inequality.

In addition, by the $L$-bounded property of $\varphi$, we also have

$$
|\varphi(\alpha z) - \varphi(0)| \leq L|\alpha z|.
$$

Hence, for any $\alpha \geq 1$, it holds that

$$
\begin{aligned}
|\varphi(\alpha z)| &\leq |\varphi(0)| + L|\alpha||z| \\
&\leq L\big(1 + |\alpha||z|\big) \\
&\leq L|\alpha|\sqrt{2(1+z^2)}.
\end{aligned}
$$

(42)

From (42), we obtain

$$
\begin{aligned}
\mathbb{E}_{a\sim\mathcal{N}(0,\alpha^2)}[\varphi^2(a)] &= \int_{-\infty}^{\infty} \frac{1}{\alpha\sqrt{2\pi}} \varphi^2(z) \exp\left(-\frac{z^2}{2\alpha^2}\right) dz \\
&= \int_{-\infty}^{\infty} \frac{1}{\sqrt{2\pi}} \varphi^2(\alpha z) \exp\left(-\frac{z^2}{2}\right) dz \\
&\leq 2L^2\alpha^2 \int_{-\infty}^{\infty} \frac{1}{\sqrt{2\pi}} (1+z^2) \exp\left(-\frac{z^2}{2}\right) dz \\
&= 4L^2\alpha^2.
\end{aligned}
\tag{43}
$$

Similarly, we also have

$$
\mathbb{E}_{b\sim\mathcal{N}(0,\beta^2)}[\varphi^2(b)] \leq 4L^2\beta^2.
\tag{44}
$$

From (40), (43) and (44), we obtain $|\tilde{Q}_{\alpha,\beta}(x)| \leq 4L^2/q^2$ for all $\alpha \geq 1$, $\beta \geq 1$, and $x \in \mathbb{R}$.

Now, for a fixed pair $(\alpha > 0, \beta > 0)$, define $z := (u,v)$, $\phi(z) := \varphi(u)\varphi(v)$, and

$$
\Sigma_x := \begin{bmatrix} \alpha^2 & x\alpha\beta \\ x\alpha\beta & \beta^2 \end{bmatrix}.
\tag{45}
$$

Then, by (Daniely et al., 2016, Lemma 12) we have

$$
\frac{\partial \tilde{Q}_{\alpha,\beta}}{\partial \Sigma_x} = -\frac{1}{2q^2\alpha\beta} \mathbb{E}_{(u,v)\sim\mathcal{N}(0,\Sigma_x)}\left[\frac{\partial\phi^2(z)}{\partial^2 z}(u,v)\right].
\tag{46}
$$

On the other hand, we note that

$$
\frac{\partial\phi^2(z)}{\partial^2 z}(u,v) = \begin{bmatrix} \frac{\partial^2\varphi(u)}{\partial u^2}\varphi(v) & \frac{\partial\varphi(u)}{\partial u}\frac{\partial\varphi(v)}{\partial v} \\ \frac{\partial\varphi(u)}{\partial u}\frac{\partial\varphi(v)}{\partial v} & \frac{\partial^2\varphi(v)}{\partial v^2}\varphi(u) \end{bmatrix}.
\tag{47}
$$

Hence, from (46) and (47) we have

$$
\left\|\frac{\partial \tilde{Q}_{\alpha,\beta}}{\partial \Sigma_x}\right\|_\infty \leq \frac{1}{2q^2\alpha\beta} \max\left\{ \mathbb{E}_{(u,v)\sim\mathcal{N}(0,\Sigma_x)}\left[\left|\frac{\partial^2\varphi(u)}{\partial u^2}\varphi(v)\right|\right], \mathbb{E}_{(u,v)\sim\mathcal{N}(0,\Sigma_x)}\left[\left|\frac{\partial\varphi(u)}{\partial u}\frac{\partial\varphi(v)}{\partial v}\right|\right], \right.
$$
$$
\left. \mathbb{E}_{(u,v)\sim\mathcal{N}(0,\Sigma_x)}\left[\left|\frac{\partial^2\varphi(v)}{\partial v^2}\sigma(u)\right|\right]\right\}.
\tag{48}
$$

Now, since $|\varphi(0)| \leq L$ and $\|\varphi'\|_\infty \leq L$, it holds that

$$
|\varphi(x)| \leq |\varphi(x) - \varphi(0)| + |\varphi(0)|
\tag{49}
$$
$$
\leq L(|x| + 1), \qquad \forall x \in \mathbb{R}.
\tag{50}
$$

Hence, by the assumption that $\|\varphi''\|_\infty \leq L$, from (48) and (50), we obtain

$$
\left\|\frac{\partial \tilde{Q}_{\alpha,\beta}}{\partial \Sigma_x}\right\|_\infty \leq \frac{L^2}{2q^2\alpha\beta} \max\left\{ \mathbb{E}_{(u,v)\sim\mathcal{N}(0,\Sigma_x)}[|u|+1], 1, \mathbb{E}_{(u,v)\sim\mathcal{N}(0,\Sigma_x)}[|v|+1]\right\}
\tag{51}
$$
$$
\leq \frac{L^2}{2q^2\alpha\beta} \max\{\alpha+1, \beta+1\}.
\tag{52}
$$

It follows that

$$\left|\tilde{Q}_{\alpha,\beta}(y) - \tilde{Q}_{\alpha,\beta}(x)\right| = \left|\int_x^y \frac{d\tilde{Q}_{\alpha,\beta}}{dt}dt\right| \tag{53}$$

$$= \left|\int_x^y \mathrm{tr}\left(\left(\frac{\partial\tilde{Q}_{\alpha,\beta}}{\partial\Sigma_t}\right)^{\mathrm{T}}\frac{\partial\Sigma_t}{dt}\right)dt\right| \tag{54}$$

$$\leq \int_x^y \left|\mathrm{tr}\left(\left(\frac{\partial\tilde{Q}_{\alpha,\beta}}{\partial\Sigma_t}\right)^{\mathrm{T}}\frac{\partial\Sigma_t}{dt}\right)\right|dt \tag{55}$$

$$\leq 4\int_x^y \left\|\frac{\partial\tilde{Q}_{\alpha,\beta}}{\partial\Sigma_t}\right\|_\infty \left\|\frac{\partial\Sigma_t}{dt}\right\|_\infty dt \tag{56}$$

$$\leq \frac{4L^2}{2q^2\alpha\beta}\max\{\alpha+1,\beta+1\}^2\int_x^y \left\|\frac{\partial\Sigma_t}{dt}\right\|_\infty dt \tag{57}$$

$$= \frac{2L^2}{q^2\alpha\beta}\max\{\alpha+1,\beta+1\}^2\alpha\beta|y-x| \tag{58}$$

$$= \frac{2L^2\max\{\alpha+1,\beta+1\}^2}{q^2}. \tag{59}$$

## B    Proof of Lemma 11

From (16) in Definition 6, we have

$$\nu_{ii}^{(l)} = \frac{\sigma_w^2\mathbf{K}_{ii}^{(l-1)} + d^{-1}\mathbf{x}_i^T\mathbf{x}_i}{\sigma_w^2\mathbf{K}_{ii}^{(l-1)} + 1}$$
$$= 1. \tag{60}$$

From (13) and (17) in Definition 6 and (60), we have

$$\rho_{ii}^{(l)} = \sigma_w^2\mathbf{K}_{ii}^{(l-1)} + 1. \tag{61}$$

In addition, from (14) and (16) in Definition 6 and (61), we also have

$$\rho_{ij}^{(l)}\nu_{ij}^{(l)} = \sigma_w^2\mathbf{K}_{ij}^{(l-1)} + d^{-1}\mathbf{x}_i^T\mathbf{x}_j, \qquad \forall i,j. \tag{62}$$

Replacing (17) in Definition 6 and (61) to (16) in Definition 6, we obtain for $i \neq j$,

$$|\nu_{ij}^{(l)}| = \frac{\left|\sigma_w^2\mathbf{K}_{ij}^{(l-1)} + d^{-1}\mathbf{x}_i^T\mathbf{x}_j\right|}{\sqrt{\left(\sigma_w^2\mathbf{K}_{ii}^{(l-1)} + 1\right)\left(\sigma_w^2\mathbf{K}_{jj}^{(l-1)} + 1\right)}}$$

$$= \frac{\left|2q^2\sigma_w^2\rho_{ij}^{(l-1)}Q_{ij}\left(\nu_{ij}^{(l-1)}\right) + d^{-1}\mathbf{x}_i^T\mathbf{x}_j\right|}{\sqrt{\rho_{ii}^{(l)}\rho_{jj}^{(l)}}}$$

$$= \frac{\left|Q_{ij}\left(\nu_{ij}^{(l-1)}\right)/\sqrt{Q_{ii}(1)Q_{jj}(1)}\sqrt{(2q^2\sigma_w^2\rho_{ii}^{(l-1)}Q_{ii}(1))(2q^2\sigma_w^2\rho_{jj}^{(l-1)}Q_{jj}(1))} + d^{-1}\mathbf{x}_i^T\mathbf{x}_j\right|}{\sqrt{\rho_{ii}^{(l)}\rho_{jj}^{(l)}}}$$

$$= \frac{\left|Q_{ij}\left(\nu_{ij}^{(l-1)}\right)/\sqrt{Q_{ii}(1)Q_{jj}(1)}\sqrt{(\rho_{ii}^{(l)} - 1)(\rho_{jj}^{(l)} - 1)} + d^{-1}\mathbf{x}_i^T\mathbf{x}_j\right|}{\sqrt{\rho_{ii}^{(l)}\rho_{jj}^{(l)}}}$$

$$\leq \frac{\sqrt{(\rho_{ii}^{(l)} - 1)(\rho_{jj}^{(l)} - 1)} + \left|d^{-1}\mathbf{x}_i^T\mathbf{x}_j\right|}{\sqrt{\rho_{ii}^{(l)}\rho_{jj}^{(l)}}} \tag{63}$$

$$\leq \frac{\sqrt{(\rho_{ii}^{(l)} - 1)(\rho_{jj}^{(l)} - 1)} + 1}{\sqrt{\rho_{ii}^{(l)} \rho_{jj}^{(l)}}} \tag{64}$$

$$\leq 1, \tag{65}$$

where (63) follows from Lemma 9, and (64) follows from $d^{-1}|\mathbf{x}_i^T \mathbf{x}_j| \leq d^{-1}\|\mathbf{x}_i\|_2 \|\mathbf{x}_j\|_2 = 1$.

## C   Proof of Proposition 12

Assume that $\mathbf{T}^{(l)} = [\mathbf{v}_1^{(l)}, \mathbf{v}_2^{(l)}, \cdots, \mathbf{v}_n^{(l)}]$ where $\mathbf{v}_i^{(l)} \in \mathbb{R}^m$ for all $i \in [n]$. By (1), we have

$$\mathbf{v}_i^{(l)} = \varphi\big(\mathbf{W}\mathbf{v}_i^{(l-1)} + \mathbf{U}\mathbf{x}_i\big), \qquad \forall i \in [n]. \tag{66}$$

Then, with probability at least $1 - \exp(-\Omega(m))$ we have

$$\begin{aligned}
\|\mathbf{v}_i^{(l)}\|_2 &= \big\|\varphi\big(\mathbf{W}\mathbf{v}_i^{(l-1)} + \mathbf{U}\mathbf{x}_i\big)\big\|_2 \\
&\leq \big\|\varphi\big(\mathbf{W}\mathbf{v}_i^{(l-1)} + \mathbf{U}\mathbf{x}_i\big) - \varphi(\underline{0})\big\|_2 + \|\varphi(\underline{0})\|_2 \\
&\leq L\big\|\mathbf{W}\mathbf{v}_i^{(l-1)} + \mathbf{U}\mathbf{x}_i\big\|_2 + \sqrt{m}L \\
&\leq L\big(\|\mathbf{W}\|_2\|\mathbf{v}_i^{(l-1)}\|_2 + \|\mathbf{U}\|_2\|\mathbf{x}_i\|_2\big) + \sqrt{m}L \\
&\leq 2L\sqrt{2}\sigma_w\|\mathbf{v}_i^{(l-1)}\|_2 + L\sqrt{d} + \sqrt{m}L, 
\end{aligned} \tag{67}$$

where (67) follows from Lemma 4, Lemma 18, and Assumption 1.

From (67), with probability at least $1 - \exp(-\Omega(m))$ we have

$$\|\mathbf{v}_i^{(l)}\|_2 \leq \frac{L\sqrt{d} + \sqrt{m}L}{1 - 2L\sqrt{2}\sigma_w}, \qquad \forall l \in \mathbb{Z}_+. \tag{68}$$

Hence, with probability at least $1 - \exp\big(-\Omega(m)\big)$, we have

$$\begin{aligned}
\big\|\mathbf{v}_i^{(l+1)} - \mathbf{v}_i^{(l)}\big\|_2 &= \big\|\varphi\big(\mathbf{W}\mathbf{v}_i^{(l)} + \mathbf{U}\mathbf{x}_i\big) - \varphi\big(\mathbf{W}\mathbf{v}_i^{(l-1)} + \mathbf{U}\mathbf{x}_i\big)\big\|_2 \\
&\leq L\big\|\mathbf{W}\big(\mathbf{v}_i^{(l)} - \mathbf{v}_i^{(l-1)}\big)\big\|_2 \\
&\leq L\big\|\mathbf{W}\big\|_2\big\|\mathbf{v}_i^{(l)} - \mathbf{v}_i^{(l-1)}\big\|_2 \\
&\leq 2L\sqrt{2}\sigma_w\big\|\mathbf{v}_i^{(l)} - \mathbf{v}_i^{(l-1)}\big\|_2 
\end{aligned} \tag{69} \tag{70}$$

where (69) is a consequence of the assumption that $\varphi$ is $L$-bounded, and (70) follows from Lemma 4 .

Now, with probability $1 - \exp(-\Omega(m))$, we have

$$\begin{aligned}
\big\|\mathbf{v}_i^{(1)}\big\|_2 &= \big\|\varphi(\mathbf{U}\mathbf{x}_i)\big\|_2 \\
&\leq \big\|\varphi(\mathbf{U}\mathbf{x}_i) - \varphi(\underline{0})\big\|_2 + \big\|\varphi(\underline{0})\big\|_2 \\
&\leq L\big\|\mathbf{U}\mathbf{x}_i\big\|_2 + \sqrt{m}L \\
&\leq L\big\|\mathbf{U}\big\|_2\|\mathbf{x}_i\|_2 + \sqrt{m}L \\
&\leq L\sqrt{d} + \sqrt{m}L, 
\end{aligned} \tag{71}$$

where (71) follows from Lemma 18 and Assumption 1.

Therefore, for all $l \geq 2$, it holds that

$$\begin{aligned}
\big\|\mathbf{v}_i^{(l)} - \mathbf{v}_i^{(l-1)}\big\|_2 &\leq \big(2L\sqrt{2}\sigma_w\big)^l\big\|\mathbf{v}_i^{(1)} - \mathbf{v}_i^{(0)}\big\|_2 \\
&= \big(2L\sqrt{2}\sigma_w\big)^l\big\|\mathbf{v}_i^{(1)}\big\|_2 \\
&\leq \big(2L\sqrt{2}\sigma_w\big)^l(L\sqrt{d} + \sqrt{m}L), 
\end{aligned} \tag{72}$$

where (72) follows from (71).

Then, for all $r > s$, with probability at least $1 - \exp(-\Omega(m))$, we have

$$\left\|\mathbf{v}_i^{(r)} - \mathbf{v}_i^{(s)}\right\| \leq \sum_{l=s+1}^{r} \left\|\mathbf{v}_i^{(l)} - \mathbf{v}_i^{(l-1)}\right\|_2$$

$$\leq (L\sqrt{d} + \sqrt{m}L) \sum_{l=s+1}^{r} \left(2L\sqrt{2}\sigma_w\right)^l$$

$$\leq (L\sqrt{d} + \sqrt{m}L)\left(2L\sqrt{2}\sigma_w\right)^{s+1} \frac{1}{1 - 2L\sqrt{2}\sigma_w} \to 0 \tag{73}$$

as $s \to \infty$ since $2L\sqrt{2}\sigma_w < 1$. It follows that $\{\mathbf{v}_i^{(l)}\}_{l=1}^{\infty}$ is a Cauchy sequence. Since $\mathbb{R}$ is complete, hence we have

$$\left\|\mathbf{v}_i^{(l)} - \mathbf{v}_i\right\| \to 0 \tag{74}$$

for some vector $\mathbf{v}_i$. Therefore, from (68) we have

$$\|\mathbf{v}_i\|_2 \leq \frac{L\sqrt{d} + \sqrt{m}L}{1 - 2L\sqrt{2}\sigma_w} \tag{75}$$

with probability at least $1 - \exp(-\Omega(m))$.

In addition, we have

$$\left\|\mathbf{v}_i^{(l-1)} - \mathbf{v}_i\right\| - \left\|\mathbf{v}_i^{(l)} - \mathbf{v}_i\right\| \leq \left\|\mathbf{v}_i^{(l)} - \mathbf{v}_i^{(l-1)}\right\|$$

$$\leq (L\sqrt{d} + \sqrt{m}L)\left(2L\sqrt{2}\sigma_w\right)^l, \qquad \forall l \geq 2. \tag{76}$$

From (76), with probability at least $1 - \exp(-\Omega(m))$ we have

$$\left\|\mathbf{v}_i^{(l)} - \mathbf{v}_i\right\| \leq (L\sqrt{d} + \sqrt{m}L)\left\| \sum_{k=l+1}^{\infty} \left(2L\sqrt{2}\sigma_w\right)^k \right\|$$

$$= (L\sqrt{d} + \sqrt{m}L)\frac{\left(2L\sqrt{2}\sigma_w\right)^{l+1}}{1 - 2L\sqrt{2}\sigma_w}. \tag{77}$$

Consequently, we have

$$\left|\mathbf{G}_{ij} - \mathbf{G}_{ij}^{(l)}\right| = \left|\mathbf{v}_i^T\mathbf{v}_j - \left(\mathbf{v}_i^{(l)}\right)^T\left(\mathbf{v}_j^{(l)}\right)\right|$$

$$\leq \left|\mathbf{v}_i^T\mathbf{v}_j - \mathbf{v}_i^T\left(\mathbf{v}_j^{(l)}\right)\right| + \left|\mathbf{v}_i^T\left(\mathbf{v}_j^{(l)}\right) - \left(\mathbf{v}_i^{(l)}\right)^T\left(\mathbf{v}_j^{(l)}\right)\right|$$

$$\leq \|\mathbf{v}_i\|\|\mathbf{v}_j - \mathbf{v}_j^{(l)}\| + \|\mathbf{v}_j^{(l)}\|\|\mathbf{v}_i - \mathbf{v}_i^{(l)}\|$$

$$\leq \|\mathbf{v}_i\|(L\sqrt{d} + \sqrt{m}L)\frac{\left(2L\sqrt{2}\sigma_w\right)^{l+1}}{1 - 2L\sqrt{2}\sigma_w}$$

$$+ \|\mathbf{v}_j^{(l)}\|(L\sqrt{d} + \sqrt{m}L)\frac{\left(2L\sqrt{2}\sigma_w\right)^{l+1}}{1 - 2L\sqrt{2}\sigma_w}$$

$$\leq 2(L\sqrt{d} + \sqrt{m}L)\left(\frac{L\sqrt{d} + \sqrt{m}L}{1 - 2L\sqrt{2}\sigma_w}\right)\frac{\left(2L\sqrt{2}\sigma_w\right)^{l+1}}{1 - 2L\sqrt{2}\sigma_w}, \tag{78}$$

where (78) follows from (68) and (75).

From (78) we obtain

$$\frac{1}{m}\left|\mathbf{G}_{ij} - \mathbf{G}_{ij}^{(l)}\right| \leq O\left(\left(2L\sqrt{2}\sigma_w\right)^l\right). \tag{79}$$

Finally, we obtain (26) from (79).

# D  Proof of Proposition 13

For all $i, j \in [n] \times [n]$, observe that

$$
\begin{aligned}
&\big|\mathbf{K}_{ij}^{(l+1)} - \mathbf{K}_{ij}^{(l)}\big| \\
&= 2q^2 \big|\rho_{ij}^{(l+1)} Q_{ij}(\nu_{ij}^{(l+1)}) - \rho_{ij}^{(l)} Q_{ij}(\nu_{ij}^{(l)})\big| \\
&\leq 2q^2 \big|\rho_{ij}^{(l+1)} Q_{ij}(\nu_{ij}^{(l+1)}) - \rho_{ij}^{(l+1)} Q_{ij}(\nu_{ij}^{(l)})\big| + 2q^2 \big|\rho_{ij}^{(l+1)} Q_{ij}(\nu_{ij}^{(l)}) - \rho_{ij}^{(l)} Q_{ij}(\nu_{ij}^{(l)})\big|,
\end{aligned}
\tag{80}
$$

where (80) follows from the triangle inequality.

Now, we bound each term in (80). First, from Assumption 1 and Lemma 9, we have

$$
2q^2 \sigma_w^2 Q_{ii}(1) \leq 8L^2 \sigma_w^2 < 1.
\tag{81}
$$

Therefore, from (13) we have

$$
\rho_{ii}^{(l)} = \frac{1 - (2q^2 \sigma_w^2 Q_{ii}(1))^{l+1}}{1 - 2q^2 \sigma_w^2 Q_{ii}(1)}, \qquad \forall i.
\tag{82}
$$

It follows that

$$
\big|\rho_{ii}^{(l)} - \rho_{ii}^{(l+1)}\big| \leq O\big((2q^2 \sigma_w^2 Q_{ii}(1))^l\big).
\tag{83}
$$

Hence, for $i \neq j$, we have

$$
\begin{aligned}
\big|\rho_{ij}^{(l+1)} - \rho_{ij}^{(l)}\big| &= \left|\sqrt{\rho_{ii}^{(l+1)} \rho_{jj}^{(l+1)}} - \sqrt{\rho_{ii}^{(l)} \rho_{jj}^{(l)}}\right| \\
&\leq \sqrt{\rho_{ii}^{(l+1)}} \left|\sqrt{\rho_{jj}^{(l+1)}} - \sqrt{\rho_{jj}^{(l)}}\right| + \sqrt{\rho_{jj}^{(l)}} \left|\sqrt{\rho_{ii}^{(l+1)}} - \sqrt{\rho_{ii}^{(l)}}\right| \\
&\leq O\big((2q^2 \sigma_w^2 Q_{ii}(1))^l\big) + O\big((2q^2 \sigma_w^2 Q_{jj}(1))^l\big),
\end{aligned}
\tag{84}
$$

where (84) follows from (82) and (83).

From (81) and (84), we obtain

$$
\big|\rho_{ij}^{(l)} - \rho_{ij}^{(l+1)}\big| \leq O\big((8L^2 \sigma_w^2)^l\big), \qquad \forall i, j.
\tag{85}
$$

Now, by (75), with probability at least $1 - \exp(-\Omega(m))$ we have

$$
0 \leq \frac{\mathbb{E}[\mathbf{G}_{ii}]}{m} \leq \frac{1}{m}\left(\frac{L\sqrt{d} + \sqrt{mL}}{1 - 2L\sqrt{2}\sigma_w}\right)^2 \leq \frac{4L}{(1 - 2L\sqrt{2}\sigma_w)^2}, \qquad \forall i.
\tag{86}
$$

Therefore, we have

$$
\begin{aligned}
&\big|\rho_{ij}^{(l+1)} Q_{ij}(\nu_{ij}^{(l+1)}) - \rho_{ij}^{(l+1)} Q_{ij}(\nu_{ij}^{(l)})\big| \\
&= \left|\rho_{ij}^{(l+1)} \tilde{Q}_{\sqrt{2\left(\frac{\sigma_w^2}{m}\mathbb{E}[\mathbf{G}_{ii}]+1\right)}, \sqrt{2\left(\frac{\sigma_w^2}{m}\mathbb{E}[\mathbf{G}_{jj}]+1\right)}}(\nu_{ij}^{(l+1)})\right. \\
&\qquad \left. - \rho_{ij}^{(l+1)} \tilde{Q}_{\sqrt{2\left(\frac{\sigma_w^2}{m}\mathbb{E}[\mathbf{G}_{ii}]+1\right)}, \sqrt{2\left(\frac{\sigma_w^2}{m}\mathbb{E}[\mathbf{G}_{jj}]+1\right)}}(\nu_{ij}^{(l)})\right| \\
&\leq \frac{2L^2}{q^2}\left(1 + \frac{4L}{(1 - 2L\sqrt{2}\sigma_w)^2}\right)\big|\rho_{ij}^{(l+1)}\nu_{ij}^{(l+1)} - \rho_{ij}^{(l+1)}\nu_{ij}^{(l)}\big| \\
&\leq \frac{2L^2}{q^2}\left(1 + \frac{4L}{(1 - 2L\sqrt{2}\sigma_w)^2}\right)\big|\rho_{ij}^{(l+1)}\nu_{ij}^{(l+1)} - \rho_{ij}^{(l)}\nu_{ij}^{(l)}\big|
\end{aligned}
\tag{87}
$$

$$
\begin{aligned}
&+ \frac{2L^2}{q^2}\left(1 + \frac{4L}{(1 - 2L\sqrt{2}\sigma_w)^2}\right)|\rho_{ij}^{(l)} - \rho_{ij}^{(l+1)}||\nu_{ij}^{(l)}| \\
&\leq \frac{2L^2}{q^2}\left(1 + \frac{4L}{(1 - 2L\sqrt{2}\sigma_w)^2}\right)|\rho_{ij}^{(l+1)}\nu_{ij}^{(l+1)} - \rho_{ij}^{(l)}\nu_{ij}^{(l)}| \\
&\quad + \frac{2L^2}{q^2}\left(1 + \frac{4L}{(1 - 2L\sqrt{2}\sigma_w)^2}\right)|\rho_{ij}^{(l)} - \rho_{ij}^{(l+1)}| \\
&= \frac{2L^2}{q^2}\left(1 + \frac{4L}{(1 - 2L\sqrt{2}\sigma_w)^2}\right)\sigma_w^2|\mathbf{K}_{ij}^{(l)} - \mathbf{K}_{ij}^{(l-1)}| \\
&\quad + \frac{2L^2}{q^2}\left(1 + \frac{4L}{(1 - 2L\sqrt{2}\sigma_w)^2}\right)O\big((8L^2\sigma_w^2)^l\big),
\end{aligned}
$$

$$(88)$$
$$(89)$$

where (87) follows from Lemma 9, (88) follows from Lemma 11, (89) follows from (23) in Lemma 11 and (85).

In addition, by using the fact that $|Q_{\alpha,\beta}(x)| \leq \frac{4L^2}{q^2}$ for all $\alpha \geq 1, \beta \geq 1$ in Lemma 9, we have

$$
\begin{aligned}
\big|\rho_{ij}^{(l+1)}Q_{ij}(\nu_{ij}^{(l)}) - \rho_{ij}^{(l)}Q_{ij}(\nu_{ij}^{(l)})\big| &\leq \frac{4L^2}{q^2}\big|\rho_{ij}^{(l+1)} - \rho_{ij}^{(l)}\big| \\
&= \frac{4L^2}{q^2}O\big((8L^2\sigma_w^2)^l\big),
\end{aligned}
$$

$$(90)$$

where (90) follows from (85).

From (17), (89), and (90) we have

$$
\begin{aligned}
\big|\mathbf{K}_{ij}^{(l+1)} &- \mathbf{K}_{ij}^{(l)}\big| \\
&= 2q^2\big|\rho_{ij}^{(l+1)}Q_{ij}(\nu_{ij}^{(l+1)}) - \rho_{ij}^{(l)}Q_{ij}(\nu_{ij}^{(l)})\big| \\
&\leq 2q^2\big|\rho_{ij}^{(l+1)}Q_{ij}(\nu_{ij}^{(l+1)}) - \rho_{ij}^{(l+1)}Q_{ij}(\nu_{ij}^{(l)})\big| + 2q^2\big|\rho_{ij}^{(l+1)}Q_{ij}(\nu_{ij}^{(l)}) - \rho_{ij}^{(l)}Q_{ij}(\nu_{ij}^{(l)})\big| \\
&\leq 2q^2\bigg[\frac{2L^2}{q^2}\left(1 + \frac{4L}{(1 - 2L\sqrt{2}\sigma_w)^2}\right)\sigma_w^2|\mathbf{K}_{ij}^{(l)} - \mathbf{K}_{ij}^{(l-1)}| \\
&\quad + \frac{2L^2}{q^2}\left(1 + \frac{4L}{(1 - 2L\sqrt{2}\sigma_w)^2}\right)O\big((8L^2\sigma_w^2)^l\big)\bigg] + 2q^2 \times \frac{4L^2}{q^2}O\big((8L^2\sigma_w^2)^l\big).
\end{aligned}
$$

$$(91)$$

By using induction, from (91) we have

$$
\big|\mathbf{K}_{ij}^{(l+1)} - \mathbf{K}_{ij}^{(l)}\big| = O\big((4L^2\sigma_w^2)^l\big).
$$

$$(92)$$

Since $\sigma_w^2 < 1/(8L^2)$, $\{\mathbf{K}_{ij}^{(l)}\}_{l=1}^{\infty}$ can be easily shown to be a Cauchy sequence. From the completeness of $\mathbb{R}$, it holds that

$$
\mathbf{K}_{ij}^{(l)} \to \mathbf{K}_{ij}
$$

$$(93)$$

uniformly in $i, j \in [n] \times [n]$ as $l \to \infty$ for some matrix $\mathbf{K}$. By using the triangle inequality, we have

$$
\big|\mathbf{K}_{ij}^{(l+1)} - \mathbf{K}_{ij}^{(l)}\big| \geq \big|\mathbf{K}_{ij}^{(l)} - \mathbf{K}_{ij}\big| - \big|\mathbf{K}_{ij}^{(l+1)} - \mathbf{K}_{ij}\big|.
$$

$$(94)$$

From (92) and (94), we obtain

$$
\big|\mathbf{K}_{ij}^{(l)} - \mathbf{K}_{ij}\big| = O\big((8L^2\sigma_w^2)^l\big).
$$

$$(95)$$

From (95), we obtain

$$
\big\|\mathbf{K}^{(l)} - \mathbf{K}\big\|_F = O\big(n(8L^2\sigma_w^2)^l\big).
$$

$$(96)$$

Now, by (17) and (93) we have

$$\mathbf{K}_{ij}^{(l)} = 2q^2 \rho_{ij}^{(l)} Q_{ij}(\nu_{ij}^{(l)}) \tag{97}$$

and $\mathbf{K}_{ij}^{(l)} \to \mathbf{K}_{ij}$. On the other hand, by (81) we have $2q^2\sigma_w^2 Q_{ii}(1) < 1$. It follows from (82) that

$$\rho_{ii}^{(l)} \to \frac{1}{1 - 2q^2\sigma_w^2 Q_{ii}(1)} \tag{98}$$

as $l \to \infty$. Hence, it holds that $\nu_{ij}^{(l)} \to \nu_{ij}$ uniformly in $i,j \in [n] \times [n]$.

Hence, by Lemma 11, we have

$$\nu_{ij} = \begin{cases} \frac{Q_{ij}\left(\nu_{ij}\right)/\sqrt{Q_{ii}(1)Q_{jj}(1)}\sqrt{(\rho_{ii}-1)(\rho_{jj}-1)}+d^{-1}\mathbf{x}_i^T\mathbf{x}_j}{\sqrt{\rho_{ii}\rho_{jj}}}, & i \neq j \\ 1, & i = j \end{cases}, \tag{99}$$

where

$$\rho_{ii} = \frac{1}{1 - 2q^2\sigma_w^2 Q_{ii}(1)}. \tag{100}$$

## E  Proof of Proposition 14

Define

$$\hat{\mathbf{G}}_{ij}^{(l)} := \mathbb{E}\left[\frac{1}{m}\mathbf{G}_{ij}^{(l)}\middle| \mathbf{h}_l, \mathbf{h}_l'\right]. \tag{101}$$

Then, by Lemma 10, we have

$$\begin{aligned}
\hat{\mathbf{G}}_{ij}^{(l)} &= \mathbb{E}\left[\frac{1}{m}\varphi(\mathbf{M}\mathbf{h}_l)^T\varphi(\mathbf{M}\mathbf{h}_l')\middle| \mathbf{h}_l, \mathbf{h}_l'\right] \\
&= \mathbb{E}_{\mathbf{w}\sim\mathcal{N}(0,2\mathbf{I})}\left[\varphi(\mathbf{w}^T\mathbf{h}_l)\varphi(\mathbf{w}^T\mathbf{h}_l')\right].
\end{aligned} \tag{102}$$

Let

$$\hat{\mathbf{A}}_{ij}^{(l)} := \mathbf{h}_l^T\mathbf{h}_l', \qquad \hat{\mathbf{A}}_{ii}^{(l)} := \|\mathbf{h}_l\|_2^2, \qquad \hat{\mathbf{A}}_{jj}^{(l)} := \|\mathbf{h}_l'\|_2^2, \tag{103}$$

and define

$$\hat{\nu}_{ij}^{(l)} := \frac{\hat{\mathbf{A}}_{ij}^{(l)}}{\sqrt{\hat{\mathbf{A}}_{ii}^{(l)}\hat{\mathbf{A}}_{jj}^{(l)}}}. \tag{104}$$

Then, we have

$$\begin{aligned}
\hat{\mathbf{G}}_{ij}^{(l)} &= \mathbb{E}_{(u,v)\sim\mathcal{N}\left(0,2\begin{bmatrix}\|\mathbf{h}_l\|^2 & \mathbf{h}_l^T\mathbf{h}_l' \\ \mathbf{h}_l^T\mathbf{h}_l' & \|\mathbf{h}_l'\|^2\end{bmatrix}\right)}\left[\varphi(u)\varphi(v)\right] \\
&= \mathbb{E}_{(u,v)\sim\mathcal{N}\left(0,\begin{bmatrix}1 & \frac{\mathbf{h}_l^T\mathbf{h}_l'}{\|\mathbf{h}_l\|\|\mathbf{h}_l'\|} \\ \frac{\mathbf{h}_l^T\mathbf{h}_l'}{\|\mathbf{h}_l\|\|\mathbf{h}_l'\|} & 1\end{bmatrix}\right)}\left[\varphi(\sqrt{2}\|\mathbf{h}_l\|u)\varphi(\sqrt{2}\|\mathbf{h}_l'\|v)\right] \\
&= 2q^2\|\mathbf{h}_l\|\|\mathbf{h}_l'\|\tilde{Q}_{\sqrt{2}\|\mathbf{h}_l\|,\sqrt{2}\|\mathbf{h}_l'\|}(\hat{\nu}_{ij}^{(l)}) \\
&= 2q^2\sqrt{\hat{\mathbf{A}}_{ii}^{(l)}\hat{\mathbf{A}}_{jj}^{(l)}}\tilde{Q}_{\sqrt{2}\|\mathbf{h}_l\|,\sqrt{2}\|\mathbf{h}_l'\|}(\hat{\nu}_{ij}^{(l)}).
\end{aligned} \tag{105}$$

Now, we consider two cases:

- **Case 1:** $i = j$.

By Lemma 10, we have

$$\mathbf{G}_{ii}^{(l+1)} = \varphi(\mathbf{Mh}_{l+1})^T \varphi(\mathbf{Mh}_{l+1}), \tag{106}$$

where

$$\|\mathbf{h}_{l+1}\|^2 = \frac{\sigma_w^2}{m}\mathbf{G}_{ii}^{(l)} + 1. \tag{107}$$

Now, for a fixed $\mathbf{h}_{l+1}$, by Beinstein's inequality and (106), it holds with probability $1 - \exp(-\Omega(m\varepsilon^2))$ that

$$\left| \frac{1}{m}\mathbf{G}_{ii}^{(l+1)} - \hat{\mathbf{G}}_{ii}^{(l+1)} \right| \leq \varepsilon/2. \tag{108}$$

On the other hand, by (68) with probability at least $1 - \exp(-\Omega(m))$ we have

$$0 \leq \frac{\mathbf{G}_{ii}^{(l)}}{m} \leq \frac{1}{m}\left( \frac{L\sqrt{d} + \sqrt{mL}}{1 - 2L\sqrt{2}\sigma_w} \right)^2 \leq \frac{4L}{(1 - 2L\sqrt{2}\sigma_w)^2}, \qquad \forall l \in \mathbb{Z}_+, i \in [n]. \tag{109}$$

Therefore, by combining (109) with (107) we have

$$1 \leq \|\mathbf{h}_{l+1}\|^2 \leq \frac{4L\sigma_w^2}{(1 - 2L\sqrt{2}\sigma_w)^2} + 1. \tag{110}$$

This means that the $\varepsilon$-net size for $\mathbf{h}_{l+1}$ is at most $\exp\left\{ O\left(l \log \frac{1}{\varepsilon}\right) \right\}$. Hence, with probability at least $1 - n^2 \exp\left( -\Omega(m\varepsilon^2) + O(l \log \frac{1}{\varepsilon}) \right)$ we have

$$\left| \frac{1}{m}\mathbf{G}_{ii}^{(l+1)} - \hat{\mathbf{G}}_{ii}^{(l+1)} \right| \leq \varepsilon/2. \tag{111}$$

Now, observe that

$$\begin{aligned}
\hat{\mathbf{G}}_{ii}^{(l+1)} &= \mathbb{E}_{\mathbf{w} \sim \mathcal{N}(0, 2\mathbf{I})}\left[ \varphi^2(\mathbf{w}^T\mathbf{h}_{l+1}) \right] \\
&= \mathbb{E}_{u \sim \mathcal{N}(0, 2\|\mathbf{h}_{l+1}\|_2^2)}[\varphi^2(u)] \\
&= \mathbb{E}_{u \sim \mathcal{N}(0,1)}[\varphi^2(\sqrt{2}\|\mathbf{h}_{l+1}\|_2 u)] \\
&= 2q^2\|\mathbf{h}_{l+1}\|_2^2 \tilde{Q}_{\sqrt{2}\|\mathbf{h}_{l+1}\|, \sqrt{2}\|\mathbf{h}_{l+1}\|}(1).
\end{aligned} \tag{112}$$

On the other hand, we also have

$$\begin{aligned}
\mathbf{K}_{ii}^{(l+1)} &= 2q^2\rho_{ii}^{(l+1)}Q_{ii}(1) \\
&= 2q^2(\sigma_w^2\mathbf{K}_{ii}^{(l)} + 1)Q_{ii}(1),
\end{aligned} \tag{113}$$

where (113) follows from (17) and Lemma 11, and (113) follows from Lemma 11.

It follows that

$$
\left| \hat{\mathbf{G}}_{ii}^{(l+1)} - \mathbf{K}_{ii}^{(l+1)} \right|
$$

$$
= 2q^2 \left| \|\mathbf{h}_{l+1}\|_2^2 \tilde{Q}_{\sqrt{2}\|\mathbf{h}_{l+1}\|_2, \sqrt{2}\|\mathbf{h}_{l+1}\|_2}(1) - \left( \sigma_w^2 \mathbf{K}_{ii}^{(l)} + 1 \right) Q_{ii}(1) \right|
$$

$$
= 2q^2 \left| \left( \frac{\sigma_w^2}{m} \mathbf{G}_{ii}^{(l)} + 1 \right) \tilde{Q}_{\sqrt{2}\|\mathbf{h}_{l+1}\|_2, \sqrt{2}\|\mathbf{h}_{l+1}\|_2}(1) - \left( \sigma_w^2 \mathbf{K}_{ii}^{(l)} + 1 \right) Q_{ii}(1) \right|
$$

$$
\leq 2q^2 \left| \left( \frac{\sigma_w^2}{m} \mathbf{G}_{ii}^{(l)} + 1 \right) \tilde{Q}_{\sqrt{2}\|\mathbf{h}_{l+1}\|_2, \sqrt{2}\|\mathbf{h}_{l+1}\|_2}(1) - \left( \sigma_w^2 \mathbf{K}_{ii}^{(l)} + 1 \right) \tilde{Q}_{\sqrt{2}\|\mathbf{h}_{l+1}\|, \sqrt{2}\|\mathbf{h}_{l+1}\|}(1) \right|
$$

$$
+ 2q^2 \left( \sigma_w^2 \mathbf{K}_{ii}^{(l)} + 1 \right) \left| \tilde{Q}_{\sqrt{2}\|\mathbf{h}_{l+1}\|_2, \sqrt{2}\|\mathbf{h}_{l+1}\|_2}(1) - Q_{ii}(1) \right|
$$

$$
\leq 2q^2 \sigma_w^2 \left| \frac{\mathbf{G}_{ii}^{(l)}}{m} - \mathbf{K}_{ii}^{(l)} \right| \left| \tilde{Q}_{\sqrt{2}\|\mathbf{h}_{l+1}\|_2, \sqrt{2}\|\mathbf{h}_{l+1}\|_2}(1) \right|
$$

$$
+ 2q^2 \left( \sigma_w^2 \mathbf{K}_{ii}^{(l)} + 1 \right) \left| \tilde{Q}_{\sqrt{2}\|\mathbf{h}_{l+1}\|_2, \sqrt{2}\|\mathbf{h}_{l+1}\|_2}(1) - Q_{ii}(1) \right|
$$

$$
\leq 8L^2 \sigma_w^2 \left| \frac{\mathbf{G}_{ii}^{(l)}}{m} - \mathbf{K}_{ii}^{(l)} \right| + 2q^2 \left( \sigma_w^2 \mathbf{K}_{ii}^{(l)} + 1 \right) \left| \tilde{Q}_{\sqrt{2}\|\mathbf{h}_{l+1}\|, \sqrt{2}\|\mathbf{h}_{l+1}\|}(1) - Q_{ii}(1) \right|, \tag{114}
$$

where (114) follows from Lemma 9.

Now, let

$$
\|\mathbf{h}\|_2^2 := \frac{\sigma_w^2}{m} \mathbf{G}_{ii} + 1. \tag{115}
$$

Then, we have

$$
\left| \|\mathbf{h}_{l+1}\|_2^2 - \|\mathbf{h}\|_2^2 \right| = \frac{\sigma_w^2}{m} \left| \mathbf{G}_{ii}^{(l)} - \mathbf{G}_{ii} \right| \tag{116}
$$

$$
= O\left( \left( 2L\sqrt{2}\sigma_w \right)^l \right) \tag{117}
$$

where (116) follows from (107) and (115), and (117) follows from (79).

Since $\|\mathbf{h}\| \geq 1$ and $\|\mathbf{h}_{l+1}\| \geq 1$, from (117) we obtain

$$
\left| \|\mathbf{h}_{l+1}\| - \|\mathbf{h}\| \right| = O\left( n\left( 2L\sqrt{2}\sigma_w \right)^l \right). \tag{118}
$$

On the other hand, by (79) it holds with probability at least $1 - \exp(-\Omega(m) - \Omega(m\varepsilon^2))$ that

$$
\left| \frac{1}{m} \mathbf{G}_{ii}^{(l+1)} - \mathbf{G}_{ii} \right| = O\left( \left( 2L\sqrt{2}\sigma_w \right)^{l+1} \right). \tag{119}
$$

Hence, from (108) and (119) with probability at least $1 - \exp(-\Omega(m) - \Omega(m\varepsilon^2))$, we have

$$
\left| \frac{\mathbf{G}_{ii}}{m} - \mathbb{E}\left[ \frac{\mathbf{G}_{ii}^{(l+1)}}{m} \middle| \mathbf{h}_{l+1} \right] \right| \leq \frac{\varepsilon}{2} + O\left( \left( 2L\sqrt{2}\sigma_w \right)^{l+1} \right) \tag{120}
$$

for any fixed $\mathbf{h}_{l+1}$. Now, observe that

$$
\mathbb{E}\left[ \frac{\mathbf{G}_{ii}^{(l+1)}}{m} \middle| \mathbf{h}_{l+1} \right] = \mathbb{E}\left[ \frac{\left\| \varphi(\mathbf{M}\mathbf{h}_{l+1}) \right\|^2}{m} \middle| \mathbf{h}_{l+1} \right]
$$

is a fixed function of $\mathbf{h}_{l+1}$. Hence, there exists, $\hat{\mathbf{h}}_{l+1}$ such that

$$\hat{\mathbf{h}}_{l+1} = \arg\max_{\mathbf{h}_{l+1}} \left| \frac{\mathbf{G}_{ii}}{m} - \mathbb{E}\left[\frac{\mathbf{G}_{ii}^{(l+1)}}{m}\Big|\mathbf{h}_{l+1}\right] \right|. \tag{121}$$

Then, with probability at least $1 - \exp(-\Omega(m) - \Omega(m\varepsilon^2))$ it holds that

$$\left| \frac{\mathbf{G}_{ii}}{m} - \mathbb{E}\left[\frac{\mathbf{G}_{ii}^{(l+1)}}{m}\right] \right| \leq \mathbb{E}\left[ \left| \frac{\mathbf{G}_{ii}}{m} - \mathbb{E}\left[\frac{\mathbf{G}_{ii}^{(l+1)}}{m}\Big|\mathbf{h}_{l+1}\right] \right| \right]$$

$$\leq \left| \frac{\mathbf{G}_{ii}}{m} - \mathbb{E}\left[\frac{\mathbf{G}_{ii}^{(l+1)}}{m}\Big|\hat{\mathbf{h}}_{l+1}\right] \right|$$

$$\leq \frac{\varepsilon}{2} + O\left( (2L\sqrt{2}\sigma_w)^{l+1} \right). \tag{122}$$

Hence, by taking $l \to \infty$, with probability $1 - \exp(-\Omega(m) - \Omega(m\varepsilon^2))$, it holds that

$$\left| \|\mathbf{h}\|^2 - \mathbb{E}[\|\mathbf{h}\|^2] \right| \leq \varepsilon, \tag{123}$$

$$\left| \|\mathbf{h}\| - \mathbb{E}[\|\mathbf{h}\|] \right| \leq \varepsilon, \tag{124}$$

where (124) follows from (123) and the fact that $\|\mathbf{h}\| \geq 1$ for any $\mathbf{h}$.

From (117), (118), (123), and (124), with probability at least $1 - \exp(-\Omega(m) - \Omega(m\varepsilon^2))$ it holds that

$$\left| \|\mathbf{h}_{l+1}\|^2 - \mathbb{E}[\|\mathbf{h}\|^2] \right| = \varepsilon + O\left( (2L\sqrt{2}\sigma_w)^l \right), \tag{125}$$

$$\left| \|\mathbf{h}_{l+1}\| - \mathbb{E}[\|\mathbf{h}\|] \right| = \varepsilon + O\left( (2L\sqrt{2}\sigma_w)^l \right). \tag{126}$$

Now, for any $a \in \mathbb{R}$ note that

$$\left| \varphi^2(\sqrt{2}\|\mathbf{h}_{l+1}\|a) - \varphi^2(\sqrt{2}\|\mathbf{h}\|a) \right|$$

$$= \left| \varphi(\sqrt{2}\|\mathbf{h}_{l+1}\|a) - \varphi(\sqrt{2}\|\mathbf{h}\|a) \right| \left| \sigma(\sqrt{2}\|\mathbf{h}_{l+1}\|a) + \sigma(\sqrt{2}\|\mathbf{h}\|a) \right|. \tag{127}$$

On the other hand, we have

$$\left| \varphi(\sqrt{2}\|\mathbf{h}_{l+1}\|a) - \varphi(\sqrt{2}\|\mathbf{h}\|a) \right| \leq L\sqrt{2}|a| \left| \|\mathbf{h}_{l+1}\| - \|\mathbf{h}\| \right|, \tag{128}$$

$$\left| \varphi(\sqrt{2}\|\mathbf{h}_{l+1}\|a) + \varphi(\sqrt{2}\|\mathbf{h}\|a) \right| \leq \left| \varphi(\sqrt{2}\|\mathbf{h}_{l+1}\|a) - \varphi(0) \right| + \left| \varphi(\sqrt{2}\|\mathbf{h}_l\|a) - \varphi(0) \right| + 2L \tag{129}$$

$$\leq L\sqrt{2}|a| \left( \|\mathbf{h}_{l+1}\| + \|\mathbf{h}_l\| \right) + 2L, \tag{130}$$

where we use the assumption that $\varphi$ is $L$-bounded in (129) and (130).

From (127), (128), and (130), we obtain

$$\left| \varphi^2(\sqrt{2}\|\mathbf{h}_{l+1}\|a) - \varphi^2(\sqrt{2}\|\mathbf{h}\|a) \right| \leq 2L^2|a|^2 \left| \|\mathbf{h}_{l+1}\|^2 - \|\mathbf{h}\|^2 \right| + 2L^2\sqrt{2}|a| \left| \|\mathbf{h}_{l+1}\| - \|\mathbf{h}\| \right|$$

$$= \left( 2L^2|a|^2 + 2L^2\sqrt{2}|a| \right) \left[ \varepsilon + O\left( (2L\sqrt{2}\sigma_w)^l \right) \right], \tag{131}$$

where (131) follows from (117) and (118).

From (131), we obtain

$$\left| \mathbb{E}_{a\sim\mathcal{N}(0,1)} \left[ \varphi^2(\sqrt{2}\|\mathbf{h}_{l+1}\|a) \right] - \mathbb{E}_{a\sim\mathcal{N}(0,1)} \left[ \varphi^2(\sqrt{2}\|\mathbf{h}\|a) \right] \right|$$

$$\leq 2L^2\sqrt{2}\mathbb{E}_{a\sim\mathcal{N}(0,1)}[|a|^2 + \sqrt{2}|a|]\left[\varepsilon + O\left((2L\sqrt{2}\sigma_w)^l\right)\right]$$

$$= 2L^2\sqrt{2}O\left(\varepsilon + (2L\sqrt{2}\sigma_w)^l\right). \tag{132}$$

Similarly, by the assumption that $\varphi$ is $L$-bounded, we also have

$$\mathbb{E}_{a\sim\mathcal{N}(0,1)} \left[ \varphi^2(\sqrt{2}\mathbb{E}[\|\mathbf{h}\|]a) \right] = O(1). \tag{133}$$

It follows that

$$\left| \tilde{Q}_{\sqrt{2}\|\mathbf{h}_{l+1}\|,\sqrt{2}\|\mathbf{h}_{l+1}\|}(1) - Q_{ii}(1) \right|$$

$$= \left| \frac{1}{2q^2\|\mathbf{h}_{l+1}\|^2} \mathbb{E}_{a\sim\mathcal{N}(0,1)} \left[ \varphi^2(\sqrt{2}\|\mathbf{h}_{l+1}\|a) \right] - \frac{1}{2q^2\mathbb{E}[\|\mathbf{h}\|^2]} \mathbb{E}_{a\sim\mathcal{N}(0,1)} \left[ \varphi^2(\sqrt{2}\mathbb{E}[\|\mathbf{h}\|]a) \right] \right|$$

$$\leq \left| \frac{1}{2q^2\|\mathbf{h}_{l+1}\|^2} \mathbb{E}_{a\sim\mathcal{N}(0,1)} \left[ \varphi^2(\sqrt{2}\|\mathbf{h}_{l+1}\|a) \right] - \frac{1}{2q^2\|\mathbf{h}_{l+1}\|^2} \mathbb{E}_{a\sim\mathcal{N}(0,1)} \left[ \varphi^2(\sqrt{2}\mathbb{E}[\|\mathbf{h}\|]a) \right] \right|$$

$$+ \left| \frac{1}{2q^2\|\mathbf{h}_{l+1}\|^2} \mathbb{E}_{a\sim\mathcal{N}(0,1)} \left[ \varphi^2(\sqrt{2}\mathbb{E}[\|\mathbf{h}\|]a) \right] - \frac{1}{2q^2\mathbb{E}[\|\mathbf{h}\|^2]} \mathbb{E}_{a\sim\mathcal{N}(0,1)} \left[ \varphi^2(\sqrt{2}\mathbb{E}[\|\mathbf{h}\|]a) \right] \right|$$

$$\leq \frac{1}{2q^2\|\mathbf{h}_{l+1}\|^2} \left| \mathbb{E}_{a\sim\mathcal{N}(0,1)} \left[ \varphi^2(\sqrt{2}\|\mathbf{h}_{l+1}\|a) \right] - \mathbb{E}_{a\sim\mathcal{N}(0,1)} \left[ \varphi^2(\sqrt{2}\mathbb{E}[\|\mathbf{h}\|]a) \right] \right|$$

$$+ \frac{1}{2q^2} \left| \frac{1}{\|\mathbf{h}_{l+1}\|^2} - \frac{1}{\mathbb{E}[\|\mathbf{h}\|^2]} \right| \mathbb{E}_{a\sim\mathcal{N}(0,1)} \left[ \varphi^2(\sqrt{2}\mathbb{E}[\|\mathbf{h}\|]a) \right]. \tag{134}$$

By combining (117), (132), and (133), from (134), we obtain

$$\left| \tilde{Q}_{\sqrt{2}\|\mathbf{h}_{l+1}\|,\sqrt{2}\|\mathbf{h}_{l+1}\|}(1) - Q_{i,i}(1) \right| = 2L^2 O\left(\varepsilon + (2L\sqrt{2}\sigma_w)^l\right) \tag{135}$$

since $\|\mathbf{h}_{l+1}\| \geq 1$ and $\mathbb{E}[\|\mathbf{h}\|] \geq 1$.

On the other hand, by (95) and the assumption $2L\sqrt{2}\sigma_w < 1$, we have

$$\|\mathbf{K}_{ii}^{(l+1)} - \mathbf{K}_{ii}\| = O\left((2L\sqrt{2}\sigma_w)^{l+1}\right). \tag{136}$$

From (135), (136), by setting

$$\varepsilon := O\left((2L\sqrt{2}\sigma_w)^{l+1}\right) \tag{137}$$

from (114), we obtain

$$\left| \hat{\mathbf{G}}_{ii}^{(l+1)} - \mathbf{K}_{ii}^{(l+1)} \right| \leq 8L^2\sigma_w^2 \left| \frac{\mathbf{G}_{ii}^{(l)}}{m} - \mathbf{K}_{ii}^{(l)} \right| + 2L^2 O\left((2L\sqrt{2}\sigma_w)^{l+1}\right). \tag{138}$$

It follows from (111) and (138) that with probability at least $1 - \exp\left\{ -\Omega(8^l L^{2l}\sigma_w^{2l}m) + O(l^2) \right\}$,

$$\left| \frac{1}{m}\mathbf{G}_{ii}^{(l+1)} - \mathbf{K}_{ii}^{(l+1)} \right| \leq \left| \frac{1}{m}\mathbf{G}_{ii}^{(l+1)} - \hat{\mathbf{G}}_{ii}^{(l+1)} \right| + \left| \hat{\mathbf{G}}_{ii}^{(l+1)} - \mathbf{K}_{ii}^{(l+1)} \right|$$

$$\leq 8L^2\sigma_w^2 \left| \frac{1}{m}\mathbf{G}_{ii}^{(l)} - \mathbf{K}_{ii}^{(l)} \right| + 2L^2 O\left((2L\sqrt{2}\sigma_w)^{l+1}\right),$$

which implies that with probability at least $1 - l \exp\left\{ -\Omega(8^l L^{2l} \sigma_w^{2l} m) + O(l^2) \right\}$, we have

$$\left| \frac{1}{m} \mathbf{G}_{ii}^{(l)} - \mathbf{K}_{ii}^{(l)} \right| = O\left( \left( 2L\sqrt{2}\sigma_w \right)^{l+1} \right). \tag{139}$$

- **Case 2:** $i \neq j$.

For this case, let

$$\|\mathbf{h}\|^2 := \frac{\sigma_w^2}{m} \mathbf{G}_{ii} + 1, \tag{140}$$

$$\|\mathbf{h}'\|^2 := \frac{\sigma_w^2}{m} \mathbf{G}_{jj} + 1. \tag{141}$$

By (79), with probability at least $1 - \exp(-\Omega(m))$, we have

$$\frac{1}{m}\left| \mathbf{G}_{ii} - \mathbf{G}_{ii}^{(l)} \right| = O\left( \left( 2L\sqrt{2}\sigma_w \right)^l \right). \tag{142}$$

In addition, we also have

$$\|\mathbf{h}_{l+1}\|^2 = \frac{\sigma_w^2}{m} \mathbf{G}_{ii}^{(l)} + 1 \geq 1, \tag{143}$$

$$\|\mathbf{h}'_{l+1}\|^2 = \frac{\sigma_w^2}{m} \mathbf{G}_{jj}^{(l)} + 1 \geq 1. \tag{144}$$

Hence, we have

$$\begin{aligned}
\left| \|\mathbf{h}_{l+1}\| - \|\mathbf{h}\| \right| &= O\left( \left| \|\mathbf{h}_{l+1}\|^2 - \|\mathbf{h}\|^2 \right| \right) \\
&= \frac{\sigma_w^2}{m} \left\| \mathbf{G}_{ii}^{(l)} - \mathbf{G}_{ii} \right\| \\
&= O\left( \left( 2L\sqrt{2}\sigma_w \right)^l \right).
\end{aligned} \tag{145}$$

Then, it holds that

$$\begin{aligned}
&\left| \hat{\mathbf{G}}_{ij}^{(l+1)} - \mathbf{K}_{ij}^{(l+1)} \right| \\
&= 2q^2 \left| \sqrt{\hat{\mathbf{A}}_{ii}^{(l+1)} \hat{\mathbf{A}}_{jj}^{(l+1)}} \tilde{Q}_{\sqrt{2}\|\mathbf{h}_{l+1}\|, \sqrt{2}\|\mathbf{h}'_{l+1}\|} (\hat{\nu}_{ij}^{(l+1)}) - \rho_{ij}^{(l+1)} Q_{ij}(\nu_{ij}^{(l+1)}) \right| \\
&\leq 2q^2 \left| \sqrt{\hat{\mathbf{A}}_{ii}^{(l+1)} \hat{\mathbf{A}}_{jj}^{(l+1)}} \tilde{Q}_{\sqrt{2}\|\mathbf{h}_{l+1}\|, \sqrt{2}\|\mathbf{h}'_{l+1}\|} (\hat{\nu}_{ij}^{(l+1)}) - \rho_{ij}^{(l+1)} \tilde{Q}_{\sqrt{2}\|\mathbf{h}_{l+1}\|, \sqrt{2}\|\mathbf{h}'_{l+1}\|} (\nu_{ij}^{(l+1)}) \right| \\
&\quad + 2q^2 \rho_{ij}^{(l+1)} \left| \tilde{Q}_{\sqrt{2}\|\mathbf{h}_{l+1}\|, \sqrt{2}\|\mathbf{h}'_{l+1}\|} (\nu_{ij}^{(l+1)}) - Q_{ij}(\nu_{ij}^{(l+1)}) \right|.
\end{aligned} \tag{146}$$

Now, for all $|x| \leq 1$, we have

$$\begin{aligned}
\left| \tilde{Q}_{\sqrt{2}\|\mathbf{h}_{l+1}\|, \sqrt{2}\|\mathbf{h}'_{l+1}\|} (x) - Q_{ij}(x) \right| &\leq \left| \tilde{Q}_{\sqrt{2}\|\mathbf{h}_{l+1}\|, \sqrt{2}\|\mathbf{h}'_{l+1}\|} (x) - \tilde{Q}_{\sqrt{2}\mathbb{E}[\|\mathbf{h}\|], \sqrt{2}\|\mathbf{h}'_{l+1}\|} (x) \right| \\
&\quad + \left| \tilde{Q}_{\sqrt{2}\mathbb{E}[\|\mathbf{h}\|], \sqrt{2}\|\mathbf{h}'_{l+1}\|} (x) - Q_{ij}(x) \right|.
\end{aligned} \tag{147}$$

On the other hand, we have

$$
\left| \tilde{Q}_{\sqrt{2}\mathbb{E}[\|\mathbf{h}\|], \sqrt{2}\|\mathbf{h}'_{l+1}\|}(x) - Q_{ij}(x) \right|
$$

$$
= \left| \frac{1}{2q^2 \mathbb{E}[\|\mathbf{h}\|]\|\mathbf{h}'_{l+1}\|} \mathbb{E}_{(a,b)^T \sim \mathcal{N}\left(0, \begin{bmatrix} 1 & x \\ x & 1 \end{bmatrix}\right)} \varphi(\sqrt{2}\mathbb{E}[\|\mathbf{h}\|]a)\varphi(\sqrt{2}\|\mathbf{h}'_{l+1}\|b) \right.
$$

$$
\left. - \frac{1}{2q^2 \mathbb{E}[\|\mathbf{h}\|]\mathbb{E}[\|\mathbf{h}'\|]} \mathbb{E}_{(a,b)^T \sim \mathcal{N}\left(0, \begin{bmatrix} 1 & x \\ x & 1 \end{bmatrix}\right)} \varphi(\sqrt{2}\mathbb{E}[\|\mathbf{h}\|]a)\varphi(\sqrt{2}\mathbb{E}[\|\mathbf{h}'\|]b) \right|
$$

$$
\leq \left| \frac{1}{2q^2 \mathbb{E}[\|\mathbf{h}\|]\|\mathbf{h}'_{l+1}\|} \mathbb{E}_{(a,b)^T \sim \mathcal{N}\left(0, \begin{bmatrix} 1 & x \\ x & 1 \end{bmatrix}\right)} \varphi(\sqrt{2}\mathbb{E}[\|\mathbf{h}\|]a)\varphi(\sqrt{2}\|\mathbf{h}'_{l+1}\|b) \right.
$$

$$
\left. - \frac{1}{2q^2 \mathbb{E}[\|\mathbf{h}\|]\|\mathbf{h}'_{l+1}\|} \mathbb{E}_{(a,b)^T \sim \mathcal{N}\left(0, \begin{bmatrix} 1 & x \\ x & 1 \end{bmatrix}\right)} \varphi(\sqrt{2}\mathbb{E}[\|\mathbf{h}\|]a)\varphi(\sqrt{2}\mathbb{E}[\|\mathbf{h}'\|]b) \right|
$$

$$
+ \left| \frac{1}{2q^2 \mathbb{E}[\|\mathbf{h}\|]\|\mathbf{h}'_{l+1}\|} \mathbb{E}_{(a,b)^T \sim \mathcal{N}\left(0, \begin{bmatrix} 1 & x \\ x & 1 \end{bmatrix}\right)} \varphi(\sqrt{2}\mathbb{E}[\|\mathbf{h}\|]a)\varphi(\sqrt{2}\mathbb{E}[\|\mathbf{h}'\|]b) \right.
$$

$$
\left. - \frac{1}{2q^2 \mathbb{E}[\|\mathbf{h}\|]\mathbb{E}[\|\mathbf{h}'\|]} \mathbb{E}_{(a,b)^T \sim \mathcal{N}\left(0, \begin{bmatrix} 1 & x \\ x & 1 \end{bmatrix}\right)} \varphi(\sqrt{2}\mathbb{E}[\|\mathbf{h}\|]a)\varphi(\sqrt{2}\mathbb{E}[\|\mathbf{h}'\|]b) \right|
$$

$$
\leq \frac{1}{2q^2 \mathbb{E}[\|\mathbf{h}\|]\|\mathbf{h}'_{l+1}\|} \mathbb{E}_{(a,b)^T \sim \mathcal{N}\left(0, \begin{bmatrix} 1 & x \\ x & 1 \end{bmatrix}\right)} \left| \varphi(\sqrt{2}\mathbb{E}[\|\mathbf{h}\|]a)\varphi(\sqrt{2}\|\mathbf{h}'_{l+1}\|b) \right.
$$

$$
\left. - \varphi(\sqrt{2}\mathbb{E}[\|\mathbf{h}\|]a)\varphi(\sqrt{2}\mathbb{E}[\|\mathbf{h}'\|]b) \right|
$$

$$
+ \frac{1}{2q^2 \mathbb{E}[\|\mathbf{h}\|]} \left| \frac{1}{\|\mathbf{h}'_{l+1}\|} - \frac{1}{\mathbb{E}[\|\mathbf{h}'\|]} \right| \mathbb{E}_{(a,b)^T \sim \mathcal{N}\left(0, \begin{bmatrix} 1 & x \\ x & 1 \end{bmatrix}\right)} \left| \varphi(\sqrt{2}\mathbb{E}[\|\mathbf{h}\|]a)\varphi(\sqrt{2}\mathbb{E}[\|\mathbf{h}'\|]b) \right|. \tag{148}
$$

In addition, by the assumption that $\varphi$ is $L$-bounded we have

$$
|\varphi(\sqrt{2}\mathbb{E}[\|\mathbf{h}\|]a)| \leq |\varphi(\sqrt{2}\mathbb{E}[\|\mathbf{h}\|]a) - \varphi(0)| + |\varphi(0)|
$$

$$
\leq \sqrt{2}\mathbb{E}[\|\mathbf{h}\|]|a| + L \tag{149}
$$

$$
|\varphi(\sqrt{2}\mathbb{E}[\|\mathbf{h}'\|]b)| \leq \sqrt{2}\mathbb{E}[\|\mathbf{h}\|]|b| + L. \tag{150}
$$

It follows that

$$
\left| \varphi(\sqrt{2}\mathbb{E}[\|\mathbf{h}\|]a)\varphi(\sqrt{2}\|\mathbf{h}'_{l+1}\|b) - \varphi(\sqrt{2}\mathbb{E}[\|\mathbf{h}\|]a)\varphi(\sqrt{2}\mathbb{E}[\|\mathbf{h}'\|]b) \right|
$$

$$
= \left| \varphi(\sqrt{2}\mathbb{E}[\|\mathbf{h}\|]a) \right| \left| \varphi(\sqrt{2}\|\mathbf{h}'_{l+1}\|b) - \varphi(\sqrt{2}\mathbb{E}[\|\mathbf{h}'\|]b) \right|
$$

$$
\leq (\sqrt{2}\mathbb{E}[\|\mathbf{h}\|]|a| + L) \left| \varphi(\sqrt{2}\|\mathbf{h}'_{l+1}\|b) - \varphi(\sqrt{2}\mathbb{E}[\|\mathbf{h}'\|]b) \right|
$$

$$
\leq (\sqrt{2}\mathbb{E}[\|\mathbf{h}\|]|a| + L)L\sqrt{2}|b| \left| \|\mathbf{h}'_{l+1}\| - \mathbb{E}[\|\mathbf{h}'\|] \right|. \tag{151}
$$

On the other hand, by (124), with probability at least $1 - \exp(-\Omega(m) - \Omega(m\varepsilon^2))$, it holds that

$$
\left| \|\mathbf{h}'\| - \mathbb{E}[\|\mathbf{h}'\|] \right| \leq \varepsilon. \tag{152}
$$

From (145) and (152), we have

$$\big|\|\mathbf{h}'_{l+1}\| - \mathbb{E}[\|\mathbf{h}'\|]\big| \le \big|\|\mathbf{h}'_{l+1}\| - \|\mathbf{h}'\|\big| + \big|\|\mathbf{h}'\| - \mathbb{E}[\|\mathbf{h}'\|]\big| \tag{153}$$

$$\le \varepsilon + O\Big(\big(2L\sqrt{2}\sigma_w\big)^l\Big). \tag{154}$$

Now, by setting

$$\varepsilon := O\Big(\big(2\sqrt{2}\sigma_w\big)^l\Big), \tag{155}$$

from (154), we obtain

$$\big|\|\mathbf{h}'_{l+1}\| - \mathbb{E}[\|\mathbf{h}'\|]\big| = O\Big(\big(2L\sqrt{2}\sigma_w\big)^l\Big). \tag{156}$$

Similarly, we also have

$$\big|\|\mathbf{h}_{l+1}\| - \mathbb{E}[\|\mathbf{h}\|]\big| = O\Big(\big(2L\sqrt{2}\sigma_w\big)^l\Big). \tag{157}$$

From (148), (151), (156) and (157), we obtain

$$\left|\tilde{Q}_{\sqrt{2}\mathbb{E}[\|\mathbf{h}\|],\sqrt{2}\|\mathbf{h}'_{l+1}\|}(x) - Q_{ij}(x)\right| = O\Big(\big(2L\sqrt{2}\sigma_w\big)^l\Big), \qquad \forall x : |x| \le 1. \tag{158}$$

Similarly, we can prove that

$$\left|\tilde{Q}_{\sqrt{2}\|\mathbf{h}_{l+1}\|,\sqrt{2}\|\mathbf{h}'_{l+1}\|}(x) - \tilde{Q}_{\sqrt{2}\mathbb{E}[\|\mathbf{h}\|],\sqrt{2}\|\mathbf{h}'_{l+1}\|}(x)\right| = O\Big(\big(2L\sqrt{2}\sigma_w\big)^l\Big), \qquad \forall x : |x| \le 1. \tag{159}$$

From (147), (158), and (159), we obtain

$$\left|\tilde{Q}_{\sqrt{2}\|\mathbf{h}_{l+1}\|,\sqrt{2}\|\mathbf{h}'_{l+1}\|}(x) - Q_{ij}(x)\right| \le O\Big(\big(2L\sqrt{2}\sigma_w\big)^l\Big), \qquad \forall x : |x| \le 1. \tag{160}$$

Next, we aim to upper bound

$$2q^2\left|\sqrt{\hat{\mathbf{A}}^{(l+1)}_{ii}\hat{\mathbf{A}}^{(l+1)}_{jj}}\,\tilde{Q}_{\sqrt{2}\|\mathbf{h}_{l+1}\|,\sqrt{2}\|\mathbf{h}'_{l+1}\|}(\hat{\nu}^{(l+1)}_{ij}) - \rho^{(l+1)}_{ij}\tilde{Q}_{\sqrt{2}\|\mathbf{h}_{l+1}\|,\sqrt{2}\|\mathbf{h}'_{l+1}\|}(\nu^{(l+1)}_{ij})\right|.$$

Observe that with probability at least $1 - n^2\exp(-\Omega(m))$, it holds for all $l$ sufficiently large that

$$\left|\sqrt{\hat{\mathbf{A}}^{(l+1)}_{ii}\hat{\mathbf{A}}^{(l+1)}_{jj}}\,\tilde{Q}_{\sqrt{2}\|\mathbf{h}_{l+1}\|,\sqrt{2}\|\mathbf{h}'_{l+1}\|}(\hat{\nu}^{(l+1)}_{ij}) - \rho^{(l+1)}_{ij}\tilde{Q}_{\sqrt{2}\|\mathbf{h}_{l+1}\|,\sqrt{2}\|\mathbf{h}'_{l+1}\|}(\nu^{(l+1)}_{ij})\right|$$

$$\le \left|\left(\sqrt{\hat{\mathbf{A}}^{(l+1)}_{ii}\hat{\mathbf{A}}^{(l+1)}_{jj}} - \rho^{(l+1)}_{ij}\right)\tilde{Q}_{\sqrt{2}\|\mathbf{h}_{l+1}\|,\sqrt{2}\|\mathbf{h}'_{l+1}\|}(\hat{\nu}^{(l+1)}_{ij})\right|$$

$$+ \left|\rho^{(l+1)}_{ij}\left(\tilde{Q}_{\sqrt{2}\|\mathbf{h}_{l+1}\|,\sqrt{2}\|\mathbf{h}'_{l+1}\|}(\hat{\nu}^{(l+1)}_{ij}) - \tilde{Q}_{\sqrt{2}\|\mathbf{h}_{l+1}\|,\sqrt{2}\|\mathbf{h}'_{l+1}\|}(\nu^{(l+1)}_{ij})\right)\right|$$

$$\le \frac{4L^2}{q^2}\left|\sqrt{\hat{\mathbf{A}}^{(l+1)}_{ii}\hat{\mathbf{A}}^{(l+1)}_{jj}} - \rho^{(l+1)}_{ij}\right| + \rho^{(l+1)}_{ij}\frac{2L^2}{q^2}\left|\hat{\nu}^{(l+1)}_{ij} - \nu^{(l+1)}_{ij}\right| \tag{161}$$

$$\le \frac{4L^2}{q^2}\left[\left|\sqrt{\hat{\mathbf{A}}^{(l+1)}_{ii}\hat{\mathbf{A}}^{(l+1)}_{jj}} - \rho^{(l+1)}_{ij}\right| + \rho^{(l+1)}_{ij}\left|\hat{\nu}^{(l+1)}_{ij} - \nu^{(l+1)}_{ij}\right|\right], \tag{162}$$

where (161) follows from Lemma 9.

On the other hand, we have

$$
\left| \sqrt{\hat{\mathbf{A}}_{ii}^{(l+1)} \hat{\mathbf{A}}_{jj}^{(l+1)}} - \rho_{ij}^{(l+1)} \right| + \rho_{ij}^{(l+1)} \left| \hat{\nu}_{ij}^{(l+1)} - \nu_{ij}^{(l+1)} \right|
$$

$$
= \left| \sqrt{\hat{\mathbf{A}}_{ii}^{(l+1)} \hat{\mathbf{A}}_{jj}^{(l+1)}} - \rho_{ij}^{(l+1)} \right|
$$

$$
+ \left| \left( \sqrt{\hat{\mathbf{A}}_{ii}^{(l+1)} \hat{\mathbf{A}}_{jj}^{(l+1)}} + \rho_{ij}^{(l+1)} - \sqrt{\hat{\mathbf{A}}_{ii}^{(l+1)} \hat{\mathbf{A}}_{jj}^{(l+1)}} \right) \hat{\nu}_{ij}^{(l+1)} - \rho_{ij}^{(l+1)} \nu_{ij}^{(l+1)} \right|
$$

$$
\leq 2 \left| \sqrt{\hat{\mathbf{A}}_{ii}^{(l+1)} \hat{\mathbf{A}}_{jj}^{(l+1)}} - \rho_{ij}^{(l+1)} \right| + \left| \sqrt{\hat{\mathbf{A}}_{ii}^{(l+1)} \hat{\mathbf{A}}_{jj}^{(l+1)}} \hat{\nu}_{ij}^{(l+1)} - \rho_{ij}^{(l+1)} \nu_{ij}^{(l+1)} \right|, \tag{163}
$$

where (163) follows from $|\hat{\nu}_{ij}^{(l)}| \leq 1$.

On the other hand, since $\rho_{ij}^{(l+1)} = \sqrt{\rho_{ii}^{(l+1)} \rho_{jj}^{(l+1)}}$, we also have

$$
|\sqrt{\hat{\mathbf{A}}_{ii}^{(l+1)} \hat{\mathbf{A}}_{jj}^{(l+1)}} - \rho_{ij}^{(l+1)}|
$$

$$
= \left| \sqrt{\left( \frac{\sigma_w^2}{m} \mathbf{G}_{ii}^{(l)} + 1 \right) \left( \frac{\sigma_w^2}{m} \mathbf{G}_{jj}^{(l)} + 1 \right)} - \sqrt{\left( \sigma_w^2 \mathbf{K}_{ii}^{(l)} + 1 \right) \left( \sigma_w^2 \mathbf{K}_{jj}^{(l)} + 1 \right)} \right|
$$

$$
= O\left( (2L\sqrt{2}\sigma_w)^l \right). \tag{164}
$$

Moreover, note that

$$
\sqrt{\hat{\mathbf{A}}_{ii}^{(l+1)} \hat{\mathbf{A}}_{jj}^{(l+1)}} \hat{\nu}_{ij}^{(l+1)} = \hat{\mathbf{A}}_{ij}^{(l+1)}
$$

$$
= \mathbf{h}_{l+1}^T \mathbf{h}_{l+1}'
$$

$$
= \frac{\sigma_w^2}{m} \mathbf{G}_{ij}^{(l)} + \frac{1}{d} \mathbf{x}_i^T \mathbf{x}_j \tag{165}
$$

and

$$
\rho_{ij}^{(l+1)} \nu_{ij}^{(l+1)} = \nu_{ij}^{(l+1)} \sqrt{\rho_{ii}^{(l+1)} \rho_{jj}^{(l+1)}}
$$

$$
= \nu_{ij}^{(l+1)} \sqrt{\left( \sigma_w^2 \mathbf{K}_{ii}^{(l)} + 1 \right) \left( \sigma_w^2 \mathbf{K}_{jj}^{(l)} + 1 \right)}
$$

$$
= \sigma_w^2 \mathbf{K}_{ij}^{(l)} + \frac{1}{d} \mathbf{x}_i^T \mathbf{x}_j. \tag{166}
$$

Thus, it holds that

$$
\left| \sqrt{\hat{\mathbf{A}}_{ii}^{(l+1)} \hat{\mathbf{A}}_{jj}^{(l+1)}} \hat{\nu}_{ij}^{(l+1)} - \rho_{ij}^{(l+1)} \nu_{ij}^{(l+1)} \right| = \sigma_w^2 \left| \frac{1}{m} \mathbf{G}_{ij}^{(l)} - \mathbf{K}_{ij}^{(l)} \right|. \tag{167}
$$

Thus, with probability at least $1 - l \exp\left( -\Omega(m\varepsilon^2) + O\left( l \log 1/\varepsilon \right) \right)$, it holds that

$$
\left| \hat{\mathbf{G}}_{ij}^{(l+1)} - \mathbf{K}_{ij}^{(l+1)} \right| \leq 8L^2 \sigma_w^2 \left| \frac{1}{m} \mathbf{G}_{ij}^{(l)} - \mathbf{K}_{ij}^{(l)} \right| + O\left( \left( 2L\sqrt{2}\sigma_w \right)^{l+1} \right). \tag{168}
$$

On the other hand, by Lemma 10, we have

$$
\mathbf{G}_{ij}^{(l+1)} = \varphi(\mathbf{M}\mathbf{h}_{l+1})^T \varphi(\mathbf{M}\mathbf{h}_{l+1}'). \tag{169}
$$

Hence, for a fixed vector pair $\mathbf{h}_{l+1}, \mathbf{h}_{l+1}'$, by Beinstein's inequality, with probability at least $1 - \exp(-\Omega(m\varepsilon^2))$ it holds that

$$
\left| \frac{1}{m} \mathbf{G}_{ij}^{(l+1)} - \hat{\mathbf{G}}_{ij}^{(l+1)} \right| \leq \varepsilon. \tag{170}
$$

Then, by using $\varepsilon$-net arguments as in Case 1, with probability at least $1 - l \exp\left(-\Omega(m\varepsilon^2) + O(l \log 1/\varepsilon)\right)$, we have

$$\left|\frac{1}{m}\mathbf{G}_{ij}^{(l+1)} - \hat{\mathbf{G}}_{ij}^{(l+1)}\right| \leq \varepsilon. \tag{171}$$

Consequently, we have

$$\left|\frac{1}{m}\mathbf{G}_{ij}^{(l+1)} - \mathbf{K}_{ij}^{(l+1)}\right| \leq \left|\frac{1}{m}\mathbf{G}_{ij}^{(l+1)} - \hat{\mathbf{G}}_{ij}^{(l+1)}\right| + \left|\hat{\mathbf{G}}_{ij}^{(l+1)} - \mathbf{K}_{ij}^{(l+1)}\right|$$

$$\leq 2O\left((2L\sqrt{2}\sigma_w)^{l+1}\right) + 8L^2\sigma_w^2 \left|\frac{1}{m}\mathbf{G}_{ij}^{(l)} - \mathbf{K}_{ij}^{(l)}\right| \tag{172}$$

where (172) follows from (168) and (171) and the choice of $\varepsilon$ in (155).

By applying the induction argument, one can show that for $l \geq 1$, it holds with probability at least $1 - l \exp\left(-\Omega(m\varepsilon^2) + O(l \log 1/\varepsilon)\right)$, we have

$$\left|\frac{1}{m}\mathbf{G}_{ij}^{(l)} - \mathbf{K}_{ij}^{(l)}\right| \leq \frac{2O\left((2L\sqrt{2}\sigma_w)^{l+1}\right)}{1 - 8L^2\sigma_w^2}. \tag{173}$$

By the choice of $\varepsilon$ in (155), it holds that with probability at least $1 - l \exp\left\{-\Omega(8^l L^{2l}\sigma_w^{2l}m) + O(l^2)\right\}$, we have

$$\left|\frac{1}{m}\mathbf{G}_{ij}^{(l)} - \mathbf{K}_{ij}^{(l)}\right| = O\left((2L\sqrt{2}\sigma_w)^l\right). \tag{174}$$

Finally, from (139) and (175) with probability at least $1 - n^2 l \exp\left\{-\Omega(8^l L^{2l}\sigma_w^{2l}m) + O(l^2)\right\}$, it holds that

$$\left\|\frac{1}{m}\mathbf{G}^{(l)} - \mathbf{K}^{(l)}\right\|_F = O\left(n(2L\sqrt{2}\sigma_w)^l\right). \tag{175}$$

## F  Proof of Theorem 8

Since $\mathbf{U}\mathbf{x}_i$ is a Gaussian vector with zero-mean and variance depending on $\|\mathbf{x}_i\|^2$. On the other hand, by the Assumption 2, $\|\mathbf{x}_i\| = \sqrt{d}$. Hence, from $\mathbf{v}_i = \varphi(\mathbf{W}\mathbf{v}_i + \mathbf{U}\mathbf{x}_i)$, it is easy to see that $\mathbb{E}[\mathbf{G}_{ii}] = \mathbb{E}[\|\mathbf{v}_i\|^2]$ does not depend on $i \in [n]$. This means that $\mathbb{E}[\mathbf{G}_{ii}] = \mathbb{E}[\mathbf{G}_{jj}]$ for all $i, j \in [n] \times [n]$. On the other hand, we have

$$\begin{aligned}
\mathbb{E}[\|\mathbf{v}_i\|^2] &= \mathbb{E}[\|\varphi(\mathbf{W}\mathbf{v}_i + \mathbf{U}\mathbf{x}_i)\|_2^2] \\
&= \mathbb{E}[\|\varphi(\mathbf{W}\mathbf{v}_i + \mathbf{U}\mathbf{x}_i) - \varphi(0) + \varphi(0)\|_2^2] \\
&\leq 2\mathbb{E}[\|\varphi(\mathbf{W}\mathbf{v}_i + \mathbf{U}\mathbf{x}_i) - \varphi(0)\|^2] + 2\|\varphi(0)\|^2 \\
&\leq 2L^2\mathbb{E}[\|\mathbf{W}\mathbf{v}_i + \mathbf{U}\mathbf{x}_i\|^2] + 2mL^2 \\
&\leq 4L^2\left(\mathbb{E}[\|\mathbf{W}\mathbf{v}_i\|^2] + \mathbb{E}[\|\mathbf{U}\mathbf{x}_i\|^2]\right) + 2mL^2.
\end{aligned} \tag{176}$$

Now, by Assumption 1 and Assumption 2 we have

$$\begin{aligned}
\mathbb{E}[\|\mathbf{W}\mathbf{v}_i\|^2] &= \mathbb{E}\left[\mathbb{E}\left[\|\mathbf{W}\mathbf{v}_i\|^2|\mathbf{t}_i\right]\right] \\
&= 2\sigma_w^2\mathbb{E}[\|\mathbf{v}_i\|^2],
\end{aligned}$$

and

$$\mathbb{E}[\|\mathbf{U}\mathbf{x}_i\|^2] = 2\mathbb{E}[\|\mathbf{x}_i\|^2] = 2d.$$

Therefore, from (176) we obtain

$$\mathbb{E}[\|\mathbf{v}_i\|^2] \leq \frac{8L^2 d + 2mL^2}{1 - 8L^2\sigma_w^2}, \tag{177}$$

or

$$0 \leq \frac{\mathbb{E}[\mathbf{G}_{ii}]}{m} \leq \frac{8L^2 d + 2L^2}{1 - 8L^2 \sigma_w^2}, \qquad \forall i \in [m]. \tag{178}$$

It follows that $Q_{ij}(x)$ has the form $\tilde{Q}_{\alpha,\alpha}(x)$ for some $\alpha \in [2, 2(\sigma_w^2 \frac{8L^2 d + 2L^2}{1 - 8L^2 \sigma_w^2} + 1)]$.

Thanks to this fact, from Proposition 13 and the assumption on this theorem, for all $(i,j) \in [n] \times [n]$, it holds that

$$\mathbf{K}_{ij} = 2q^2 Q_{ij}(\nu_{ij}) \sqrt{\rho_{ii}\rho_{jj}}$$
$$= 2q^2 \sqrt{\rho_{ii}\rho_{jj}} \sum_{r=0}^{\infty} \mu_{r,\alpha}^2(\varphi) \nu_{ij}^r,$$

where

$$\nu_{ij} = \frac{Q_{ij}(\nu_{ij}) / \sqrt{Q_{ii}(1)Q_{jj}(1)} \sqrt{(\rho_{ii}-1)(\rho_{jj}-1)} + d^{-1}\mathbf{x}_i^T \mathbf{x}_j}{\sqrt{\rho_{ii}\rho_{jj}}}. \tag{179}$$

Here,

$$\rho_{ii} = \frac{1}{1 - 2q^2 \sigma_w^2 Q_{ii}(1)}. \tag{180}$$

Now, by Lemma 11, we have $|\nu_{ij}| \leq 1$ for all $(i,j) \in [n] \times [n]$. Let $\mathbf{H} = [\mathbf{h}_1, \mathbf{h}_2, \cdots, \mathbf{h}_n]$ where $\mathbf{h}_1, \mathbf{h}_2, \cdots, \mathbf{h}_n$ be unit vectors such that $\nu_{ij} = \mathbf{h}_i^T \mathbf{h}_j$ for all $(i,j) \in [n] \times [n]$. It is easy to check that $[(\mathbf{H}^T \mathbf{H})^{\odot r}]_{ij} = (\mathbf{h}_i^T \mathbf{h}_j)^r$ holds for all $(i,j) \in [n] \times [n]$. Let $\tilde{\mathbf{K}}$ be a $n \times n$ matrix such that

$$\tilde{\mathbf{K}}_{ij} = \mathbf{K}_{ij} / \sqrt{\rho_{ii}\rho_{jj}}, \qquad \forall i,j \in [n] \times [n]. \tag{181}$$

Then, $\tilde{\mathbf{K}}$ can be written as

$$\tilde{\mathbf{K}} = 2q^2 \sum_{r=0}^{\infty} \mu_{r,\alpha}^2(\varphi) (\mathbf{H}^T \mathbf{H})^{(\odot r)}. \tag{182}$$

Now, for any unit vector $\mathbf{u} = [u_1, u_2, \cdots, u_n]^T \in \mathbb{R}^n$, it holds that

$$\mathbf{u}^T (\mathbf{H}^T \mathbf{H})^{(\odot r)} \mathbf{u} = \sum_{i,j} u_i u_j (\mathbf{h}_i^T \mathbf{h}_j)^r$$
$$= \sum_i u_i^2 + \sum_{i \neq j} u_i u_j \nu_{ij}^r$$
$$= 1 + \sum_{i \neq j} u_i u_j \nu_{ij}^r. \tag{183}$$

Next, we show that $|\nu_{ij}| < 1$ if $i \neq j$. Indeed, assume that there exists $i \neq j$ such that $|\nu_{ij}| \geq 1$. Then, from (30) in Lemma 11, we have

$$1 \leq |\nu_{ij}|$$
$$= \left| \frac{Q_{ij}(\nu_{ij}) / \sqrt{Q_{ii}(1)Q_{jj}(1)} \sqrt{(\rho_{ii}-1)(\rho_{jj}-1)} + d^{-1}\mathbf{x}_i^T \mathbf{x}_j}{\sqrt{\rho_{ii}\rho_{jj}}} \right|$$
$$\leq \frac{\sqrt{(\rho_{ii}-1)(\rho_{jj}-1)} + |d^{-1}\mathbf{x}_i^T \mathbf{x}_j|}{\sqrt{\rho_{ii}\rho_{jj}}} \tag{184}$$
$$< \frac{\sqrt{(\rho_{ii}-1)(\rho_{jj}-1)} + 1}{\sqrt{\rho_{ii}\rho_{jj}}} \tag{185}$$
$$\leq 1, \tag{186}$$

where (184) follows from Lemma 9, and (185) follows by the fact that since $\mathbf{x}_i \nparallel \mathbf{x}_j$, from Cauchy–Schwarz inequality and Assumption 2, we have $\mathbf{x}_i^T \mathbf{x}_j < \|\mathbf{x}_i\|_2 \|\mathbf{x}_j\| = d$. This is a contradiction. Hence, we have $|\beta| < 1$ where

$$\beta := \max_{i \neq j} |\nu_{ij}|. \tag{187}$$

Now, by taking $r > -\frac{\log(2n)}{\log \beta}$, we have

$$\left| \sum_{i \neq j} u_i u_j \nu_{ij}^r \right| \leq \sum_{i \neq j} |u_i| |u_j| \beta^r$$
$$\leq \left( \sum_i |\nu_i| \right)^2 \beta^r k$$
$$\leq n\beta^r$$
$$< \frac{1}{2}. \tag{188}$$

From (183) and (188), we obtain

$$\mathbf{u}^T \left( \mathbf{H}^T \mathbf{H} \right)^{(\odot r)} \mathbf{u} > \frac{1}{2}, \qquad \forall \mathbf{u},$$

so $\left( \mathbf{H}^T \mathbf{H} \right)^{(\odot r)}$ is positive definite. Following the assumption in Theorem 8, it holds that $\min_{\alpha \in \left[ 2, 2\left( \sigma_w^2 \frac{8L^2 d + 2L^2}{1 - 8L^2 \sigma_w^2} + 1 \right) \right]} \mu_{r,\alpha}^2(\varphi) > 0$ for infinitely many values of $r$. Hence, $\tilde{\mathbf{K}}$ is positive definite for all initializations since $0 \leq \frac{\mathbb{E}[\mathbf{G}_{ii}]}{m} \leq \frac{8L^2 d + 2L^2}{1 - 8L^2 \sigma_w^2}$.

Now, let $\mathbf{\Gamma} = \{\sqrt{\rho_{ii} \rho_{jj}}\}_{i,j}$ be an $n \times n$ matrix where the $(i, j)$ element is $\sqrt{\rho_{ii} \rho_{jj}}$. Then, we have

$$\mathbf{K} = \tilde{\mathbf{K}} \odot \mathbf{\Gamma}. \tag{189}$$

Now, for any vector $\mathbf{u} = [u_1, u_2, \cdots, u_n]^T$, we have

$$\mathbf{u}^T \mathbf{\Gamma} \mathbf{u} = \sum_{i,j} u_i u_j \sqrt{\rho_{ii} \rho_{jj}}$$
$$= \left( \sum_i u_i \sqrt{\rho_{ii}} \right)^2$$
$$\geq 0. \tag{190}$$

Hence, $\mathbf{\Gamma}$ is positive semi-definite. Now, by applying (Ling et al., 2022, Lemma 6), we have

$$\lambda_{\min}(\mathbf{K}) \geq \left( \min_i \rho_{ii} \right) \lambda_{\min}(\tilde{\mathbf{K}})$$
$$\geq \lambda_{\min}(\tilde{\mathbf{K}})$$
$$\geq \min_{\frac{\mathbb{E}[\mathbf{G}_{ii}]}{m} \in \left[ 0, \frac{8L^2 d + 2L^2}{1 - 8L^2 \sigma_w^2} \right], \forall i} \lambda_{\min}(\tilde{\mathbf{K}}) := \lambda_0^* > 0,$$

so $\mathbf{K}$ is positive definite with the smallest eigenvalue $\lambda_* \geq \lambda_0^* > 0$, where $\lambda_0^*$ is some constant which does not depend on $m$.

# G   Proof of Theorem 3

The following proof follows the same steps as (Ling et al., 2022, Proof of Theorem 1). There are some changes caused by the new activation function. First, we recall the two important auxiliary lemmas:

**Lemma 21.** *(Horn & Johnson, 1985, Sect. 5.8) Let $\mathbf{\Delta} = \mathbf{B} - \mathbf{A}$ where $\mathbf{A}$ and $\mathbf{B}$ are square complex matrices. Then, it holds that*

$$\|\mathbf{B}^{-1}\| \leq \frac{\|\mathbf{A}^{-1}\|}{1 - \|\mathbf{A}^{-1}\mathbf{\Delta}\|}. \tag{191}$$

**Lemma 22.** *(Weyl's inequality)(Ling et al., 2022, Lemma 5) Let $\mathbf{A}, \mathbf{B} \in \mathbb{R}^{m \times n}$ with their singular values satisfying $\sigma_1(\mathbf{A}) \geq \sigma_2(\mathbf{A}) \geq \cdots \geq \sigma_r(\mathbf{A})$ and $\sigma_1(\mathbf{B}) \geq \sigma_2(\mathbf{B}) \geq \cdots \geq \sigma_r(\mathbf{B})$ and $r = \min(m, n)$. Then,*

$$\max_{i \in [r]} \left| \sigma_i(\mathbf{A}) - \sigma_i(\mathbf{B}) \right| \leq \|\mathbf{A} - \mathbf{B}\|. \tag{192}$$

The equilibrium point of Eq. (2) is the root of the function $F(\tau) := \mathbf{T}(\tau) - \varphi(\mathbf{W}(\tau)\mathbf{T}(\tau) + \mathbf{U}(\tau)\mathbf{X}) = 0$. Let $\mathbf{J}(\tau) := \partial\text{vec}(\mathrm{F}(\tau))/\partial\text{vec}(\mathbf{T}(\tau))$ denote the Jacobian matrix. Then, it is easy to see that

$$\mathbf{J}(\tau) = \mathbf{I}_{mn} - \mathbf{D}(\tau)\big(\mathbf{I}_n \otimes \mathbf{W}(\tau)\big),$$

where $\mathbf{D}(\tau) := \text{diag}[\text{vec}(\varphi'(\mathbf{W}(\tau)\mathbf{T}(\tau) + \mathbf{U}(\tau)\mathbf{X}))]$. Using the Lipschitz property of activation function, it is easy to check that $\mathbf{J}(\tau)$ is invertible if $\|\mathbf{W}(\tau)\| < 1/L$. The gradient of each trainable parameter is given by the following lemma.

**Lemma 23.** *(Ling et al., 2022, Lemma 2) If $\mathbf{J}(\tau)$ is invertible, the gradient of the objective function $\Phi(\tau)$ w.r.t. each trainable parameters is given by*

$$\text{vec}(\nabla_{\mathbf{W}}\Phi(\tau)) = (\mathbf{T}(\tau) \otimes \mathbf{I}_m)\mathbf{R}(\tau)^T(\hat{\mathbf{y}}(\tau) - \mathbf{y})$$
$$\text{vec}(\nabla_{\mathbf{U}}\Phi(\tau)) = (\mathbf{X} \otimes \mathbf{I}_m)\mathbf{R}(\tau)^T(\hat{\mathbf{y}}(\tau) - \mathbf{y}),$$
$$\nabla_{\mathbf{a}}\Phi(\tau) = \mathbf{T}(\tau)(\hat{\mathbf{y}}(\tau) - \mathbf{y})$$

*where $\mathbf{R}(\tau) = (\mathbf{a}(\tau) \otimes \mathbf{I}_n)\mathbf{J}(\tau)^{-1}\mathbf{D}(\tau)$.*

Based on these three lemmas, we can prove the following result:

**Lemma 24.** *For each $s \in [0, \tau]$, suppose that $\|\mathbf{W}(s)\|_2 \leq \bar{\rho}_w, \|\mathbf{U}(s)\|_2 \leq \bar{\rho}_u$ and $\|\mathbf{a}(s)\|_2 \leq \bar{\rho}_a$. It holds that*

$$\|\mathbf{T}(s)\|_F \leq c_a\|\mathbf{X}\|_F + c_m \tag{193}$$

*and*

$$\|\nabla_{\mathbf{W}}\Phi(s)\|_F \leq c_u\big(c_a\|\mathbf{X}\|_F + c_m\big)\|\hat{\mathbf{y}}(s) - \mathbf{y}\|_2, \tag{194}$$
$$\|\nabla_U\Phi(s)\|_F \leq c_u\|\mathbf{X}\|_F\|\hat{\mathbf{y}}(s) - \mathbf{y}\|_2, \tag{195}$$
$$\|\nabla_a\Phi(s)\|_F \leq \big(c_a\|\mathbf{X}\|_F + c_m\big)\|\hat{\mathbf{y}}(s) - \mathbf{y}\|_2. \tag{196}$$

*Furthermore, for each $k, s \in [0, \tau]$, it holds that*

$$\|\mathbf{T}(k) - \mathbf{T}(s)\| \leq \frac{L}{1 - L\bar{\rho}_w}\big(c_a\|\mathbf{X}\|_F + c_m\big)\|\mathbf{W}(k) - \mathbf{W}(s)\|_2$$
$$+ \frac{L}{1 - L\bar{\rho}_w}\|\mathbf{U}(k) - \mathbf{U}(s)\|_2\|\mathbf{X}\|_F \tag{197}$$

*and*

$$\|\hat{\mathbf{y}}(k) - \hat{\mathbf{y}}(s)\|_2$$
$$\leq \bar{\rho}_a\left[\frac{L}{1 - L\bar{\rho}_w}\big(c_a\|\mathbf{X}\|_F + c_m\big)\|\mathbf{W}(k) - \mathbf{W}(s)\|_2\right.$$
$$\left. + \frac{L}{1 - L\bar{\rho}_w}\|\mathbf{U}(k) - \mathbf{U}(s)\|_2\|\mathbf{X}\|_F\right] + \big(c_a\|\mathbf{X}\|_F + c_m\big)\|\mathbf{a}(k) - \mathbf{a}(s)\|_2. \tag{198}$$

*Proof.* Observe that $\mathbf{T}(s) = \varphi(\mathbf{W}(s)\mathbf{T}(s) + \mathbf{U}(s)\mathbf{X})$. Using the fact that $|\varphi(x) - \varphi(0)| \leq L|x|$ (Lipschitz condition of $\varphi$), we have

$$
\begin{aligned}
\|\mathbf{T}(s) - \varphi(0)\|_F &= \big\|\varphi(\mathbf{W}(s)\mathbf{T}(s) + \mathbf{U}(s)\mathbf{X}) - \varphi(0)\big\|_F \\
&\leq L\|\mathbf{W}(s)\mathbf{T}(s) + \mathbf{U}(s)\mathbf{X}\|_F \\
&\leq L\left(\|\mathbf{W}(s)\|_2\|\mathbf{T}(s)\|_F + \|\mathbf{U}(s)\|_2\|\mathbf{X}\|_F\right) \\
&\leq L\bar{\rho}_w\|\mathbf{T}(s)\|_F + L\bar{\rho}_u\|\mathbf{X}\|_F.
\end{aligned}
\tag{199}
$$

From (199), we have

$$
\begin{aligned}
\|\mathbf{T}(s)\|_F &\leq \|\varphi(0)\|_F + L\bar{\rho}_w\|\mathbf{T}(s)\|_F + L\bar{\rho}_u\|\mathbf{X}\|_F \\
&= |\varphi(0)|\sqrt{mn} + L\bar{\rho}_w\|\mathbf{T}(s)\|_F + L\bar{\rho}_u\|\mathbf{X}\|_F.
\end{aligned}
\tag{200}
$$

Since $\bar{\rho}_w < 1/L$, from (200), we obtain

$$
\|\mathbf{T}(s)\|_F \leq c_a\|\mathbf{X}\|_F + c_m.
\tag{201}
$$

Now, we prove (194)-(196). By using Lemma 21 with $\mathbf{A} = \mathbf{I}_{mn}, \mathbf{B} = \mathbf{J}(s), \boldsymbol{\Delta} = -\mathbf{D}(s)(\mathbf{I}_n \otimes \mathbf{W}(s))$, we have

$$
\begin{aligned}
\|\mathbf{J}(s)^{-1}\|_2 &\leq \frac{1}{1 - \|\mathbf{D}(\tau)(\mathbf{I}_n \otimes \mathbf{W}(s))\|_2} \\
&\leq \frac{1}{1 - \|\mathbf{D}(s)\|_2\|\mathbf{W}(s)\|_2}.
\end{aligned}
\tag{202}
$$

On the other hand since $\|\varphi'\|_\infty \leq L$, we have

$$
\|\mathbf{D}(s)\|_2 \leq L.
\tag{203}
$$

Hence, from (202), we have

$$
\|\mathbf{J}(s)^{-1}\|_2 \leq \frac{1}{1 - L\bar{\rho}_w},
$$

and thus it holds that

$$
\begin{aligned}
\|\mathbf{R}(s)\|_2 &\leq \|\mathbf{a}(s)\|_2\|\mathbf{J}(s)^{-1}\|_2\|\mathbf{D}(s)\|_2 \\
&\leq \frac{L\bar{\rho}_a}{1 - L\bar{\rho}_w}.
\end{aligned}
\tag{204}
$$

Then, we have

$$
\begin{aligned}
\|\nabla_{\mathbf{W}}\Phi(s)\|_F &= \|\text{vec}(\nabla_{\mathbf{W}}\Phi(s))\|_2 \\
&= \|(\mathbf{T}(s) \otimes \mathbf{I}_m)\mathbf{R}(s)^T(\hat{\mathbf{y}}(s) - \mathbf{y})\|_2 \\
&\leq \|\mathbf{T}(s)\|_2\|\mathbf{R}(s)\|_2\|\hat{\mathbf{y}}(s) - \mathbf{y}\|_2 \\
&\leq \frac{L\bar{\rho}_a}{1 - L\bar{\rho}_w}(c_a\|\mathbf{X}\|_F + c_m)\|\hat{\mathbf{y}}(s) - \mathbf{y}\|_2,
\end{aligned}
\tag{205}
$$

$$
\begin{aligned}
\|\nabla_{\mathbf{U}}\Phi(s)\|_F &= \|\text{vec}(\nabla_{\mathbf{U}}\Phi(s))\|_2 \\
&= \|(\mathbf{X} \otimes \mathbf{I}_m)\mathbf{R}(s)^T(\hat{\mathbf{y}}(s) - \mathbf{y})\|_2 \\
&\leq \frac{L\bar{\rho}_a}{1 - L\bar{\rho}_w}\|\mathbf{X}\|_F\|\hat{\mathbf{y}}(s) - \mathbf{y}\|_2,
\end{aligned}
\tag{206}
$$

$$
\begin{aligned}
\|\nabla_{\mathbf{a}}\Phi(s)\|_F &= \|\mathbf{T}(s)(\hat{\mathbf{y}}(s) - \mathbf{y})\| \\
&\leq \big(c_a\|\mathbf{X}\|_F + c_m\big)\|\hat{\mathbf{y}}(s) - \mathbf{y}\|_2.
\end{aligned}
\tag{207}
$$

Next, we prove (197). Observe that

$$
\begin{aligned}
\|\mathbf{T}(k) &- \mathbf{T}(s)\|_F \\
&= \|\varphi(\mathbf{W}(k)\mathbf{T}(k) + \mathbf{U}(k)\mathbf{X}) - \varphi(\mathbf{W}(s)\mathbf{T}(s) + \mathbf{U}(s)\mathbf{X})\|_F \\
&\leq L\|\mathbf{W}(k)\mathbf{T}(k) + \mathbf{U}(k)\mathbf{X} - \mathbf{W}(s)\mathbf{T}(s) - \mathbf{U}(s)\mathbf{X}\|_F \\
&\leq L\big(\|\mathbf{W}(k)\mathbf{T}(k) - \mathbf{W}(k)\mathbf{T}(s)\|_F + \|\mathbf{W}(k)\mathbf{T}(s) - \mathbf{W}(s)\mathbf{T}(s)\|_F \\
&\quad + \|\mathbf{U}(k)\mathbf{X} - \mathbf{U}(s)\mathbf{X}\|_F\big) \\
&\leq L\|\mathbf{W}(k)\|_2\|\mathbf{T}(k) - \mathbf{T}(s)\|_F + L\|\mathbf{W}(k) - \mathbf{W}(s)\|_2\|\mathbf{T}(s)\|_F \\
&\quad + L\|\mathbf{U}(k) - \mathbf{U}(s)\|_2\|\mathbf{X}\|_F \\
&\leq L\bar{\rho}_w\|\mathbf{T}(k) - \mathbf{T}(s)\|_F + L\big(c_a\|\mathbf{X}\|_F + c_m\big)\|\mathbf{W}(k) - \mathbf{W}(s)\|_2 \\
&\quad + L\|\mathbf{U}(k) - \mathbf{U}(s)\|_2\|\mathbf{X}\|_F.
\end{aligned}
\tag{208}
$$

From (208), we obtain

$$
\begin{aligned}
\|\mathbf{T}(k) - \mathbf{T}(s)\|_F &\leq \frac{L}{1 - L\bar{\rho}_w}\big(c_a\|\mathbf{X}\|_F + c_m\big)\|\mathbf{W}(k) - \mathbf{W}(s)\|_2 \\
&\quad + \frac{L}{1 - L\bar{\rho}_w}\|\mathbf{U}(k) - \mathbf{U}(s)\|_2\|\mathbf{X}\|_F.
\end{aligned}
\tag{209}
$$

Finally, we prove (198). Observe that

$$
\begin{aligned}
\|\hat{\mathbf{y}}(k) &- \hat{\mathbf{y}}(s)\|_F \\
&= \|\mathbf{a}(k)\mathbf{T}(k) - \mathbf{a}(s)\mathbf{Z}(s)\|_F \\
&\leq \|\mathbf{a}(k)\mathbf{T}(k) - \mathbf{a}(k)\mathbf{T}(s)\|_F + \|\mathbf{a}(k)\mathbf{T}(s) - \mathbf{a}(s)\mathbf{T}(s)\|_F \\
&\leq \|\mathbf{a}(k)\|_2\|\mathbf{T}(k) - \mathbf{T}(s)\|_F + \|\mathbf{a}(k) - \mathbf{a}(s)\|_2\|\mathbf{T}(s)\|_F \\
&\leq \bar{\rho}_a\bigg[\frac{L}{1 - L\bar{\rho}_w}\big(c_a\|\mathbf{X}\|_F + c_m\big)\|\mathbf{W}(k) - \mathbf{W}(s)\|_2 \\
&\quad + \frac{L}{1 - L\bar{\rho}_w}\|\mathbf{U}(k) - \mathbf{U}(s)\|_2\|\mathbf{X}\|_F\bigg] + \big(c_a\|\mathbf{X}\|_F + c_m\big)\|\mathbf{a}(k) - \mathbf{a}(s)\|_2.
\end{aligned}
\tag{210}
$$

$\square$

Next, we prove the following result.

**Lemma 25.** *Under the condition $\|\mathbf{W}(\tau)\|_2 < 1/L$, the recursion in (1), given by*

$$
\mathbf{T}^{(l)} = \varphi(\mathbf{W}(\tau)\mathbf{T}^{(l-1)} + \mathbf{U}(\tau)\mathbf{X})
$$

*converges. This implies that $\mathbf{T}^{(l)} \to \mathbf{T}$ for some $\mathbf{T} \in \mathbb{R}^{m \times n}$, meaning the equilibrium point always exists.*

*Proof.* Define

$$
\alpha_\tau := L\|\mathbf{W}(\tau)\|_2 < 1.
\tag{211}
$$

Observe that

$$
\begin{aligned}
\|\mathbf{T}^{(l+1)} - \mathbf{T}^{(l)}\|_F &= \|\varphi(\mathbf{W}(\tau)\mathbf{T}^{(l)} + \mathbf{U}(\tau)\mathbf{X}) - \varphi(\mathbf{W}(\tau)\mathbf{T}^{(l-1)} + \mathbf{U}(\tau)\mathbf{X})\| \\
&\leq L\|\mathbf{W}(\tau)(\mathbf{T}^{(l)} - \mathbf{T}^{(l-1)})\|_F \tag{212} \\
&\leq L\|\mathbf{W}(\tau)\|_2\|\mathbf{T}^{(l)} - \mathbf{T}^{(l-1)}\|_F \\
&= \alpha_\tau\|\mathbf{T}^{(l)} - \mathbf{T}^{(l-1)}\|_F, \tag{213}
\end{aligned}
$$

where (212) follows from the assumption that $\varphi$ is $L$-bounded.

It follows from (213) that for all $l \in \mathbb{Z}_+$,

$$
\begin{aligned}
\left\| \mathbf{T}^{(l)} - \mathbf{T}^{(l-1)} \right\|_F &\leq (\alpha_\tau)^{l-1} \left\| \mathbf{T}^{(1)} - \mathbf{T}^{(0)} \right\|_F \\
&= (\alpha_\tau)^{l-1} \left\| \mathbf{T}^{(1)} \right\|_F \\
&= (\alpha_\tau)^{l-1} \left\| \varphi(\mathbf{U}(\tau)\mathbf{X}) \right\|_F \\
&\leq (\alpha_\tau)^{l-1} \left( \left\| \varphi(\mathbf{U}(\tau)\mathbf{X}) - \varphi(\underline{0}) \right\|_F + \left\| \varphi(\underline{0}) \right\|_F \right) \\
&\leq (\alpha_\tau)^{l-1} \left( L \| \mathbf{U}(\tau)\mathbf{X} \|_F + \| \varphi(0) \|_F \right) \tag{214} \\
&\leq (\alpha_\tau)^{l-1} \left( L \| \mathbf{U}(\tau) \|_2 \| \mathbf{X} \|_F + L\sqrt{mn} \right), \tag{215}
\end{aligned}
$$

where (214) follows from the $L$-boundedness of $\varphi$.

From (215) for all $p, q \in \mathbb{Z}_+, p > q$, we have

$$
\begin{aligned}
\left\| \mathbf{T}^{(p)} - \mathbf{T}^{(q)} \right\|_F &\leq \sum_{l=q+1}^{p} \left\| \mathbf{T}^{(l)} - \mathbf{T}^{(l-1)} \right\|_F \\
&\leq \sum_{l=q+1}^{p} (\alpha_\tau)^{l-1} \left( L \| \mathbf{U}(\tau) \|_2 \| \mathbf{X} \|_F + L\sqrt{mn} \right) \tag{216} \\
&\leq \left( L \| \mathbf{U}(\tau) \|_2 \| \mathbf{X} \|_F + L\sqrt{mn} \right) (\alpha_\tau)^q \frac{1}{1 - \alpha_\tau}, \tag{217}
\end{aligned}
$$

which can be arbitrary small for $q$ sufficiently large. Hence, $\{\mathbf{T}^{(l)}\}_{l=1}^{\infty}$ is a Cauchy sequence in $\mathbb{R}^{m \times n}$ (a Banach space). Therefore, $\mathbf{T}^{(l)}$ converges, and the equilibrium point always exists. $\qquad \square$

Now, we return to prove Theorem 3. We prove by induction for every $\tau > 0$,

$$
\| \mathbf{W}(s) \| \leq \bar{\rho}_w, \| \mathbf{U}(s) \| \leq \bar{\rho}_u, \| \mathbf{a}(s) \|_2 \leq \bar{\rho}_a, s \in [0, \tau], \tag{218}
$$

$$
\lambda_s \geq \frac{\lambda_0}{2}, s \in [0, \tau], \tag{219}
$$

$$
\Phi(s) \leq \left( 1 - \eta \frac{\lambda_0}{2} \right)^s \Phi(0), \qquad s \in [0, \tau]. \tag{220}
$$

For $\tau = 0$, it is clear that (218)-(220) hold. Assume that (218)-(220) holds up to $\tau$ iterations. Then, by using triangle inequality, we have

$$
\begin{aligned}
\| \mathbf{W}(\tau + 1) - \mathbf{W}(0) \|_F &\leq \sum_{s=0}^{\tau} \| \mathbf{W}(s+1) - \mathbf{W}(s) \|_F \\
&= \sum_{s=0}^{\tau} \eta \| \nabla_{\mathbf{W}} \Phi(s) \|_F \\
&\leq \eta \sum_{s=0}^{\tau} c_u \left( c_a \| \mathbf{X} \|_F + c_m \right) \| \hat{\mathbf{y}}(s) - \mathbf{y} \|_2 \tag{221} \\
&= \eta c_u \left( c_a \| \mathbf{X} \|_F + c_m \right) \sum_{s=0}^{\tau} \left( 1 - \eta \frac{\lambda_0}{2} \right)^{s/2} \| \hat{\mathbf{y}}(0) - \hat{\mathbf{y}} \|_2, \tag{222}
\end{aligned}
$$

where (221) follows from Lemma 24. Let $u := \sqrt{1 - \eta \lambda_0 / 2}$. Then $\| \mathbf{W}(\tau+1) - \mathbf{W}(0) \|_F$ can be bounded with

$$
\begin{aligned}
&\frac{2}{\lambda_0} (1 - u^2) \frac{1 - u^{\tau+1}}{1 - u} c_u \left( c_a \| \mathbf{X} \|_F + c_m \right) \| \hat{\mathbf{y}}(0) - \mathbf{y} \| \\
&\leq \frac{4}{\lambda_0} c_u \left( c_a \| \mathbf{X} \|_F + c_m \right) \| \hat{\mathbf{y}}(0) - \mathbf{y} \| \\
&\leq \delta. \tag{223}
\end{aligned}
$$

Then, we have

$$\|\mathbf{W}(\tau+1)\| \le \|\mathbf{W}(0)\|_2 + \delta = \bar{\rho}_w < 1/L. \tag{224}$$

Using the similar technique, one can show that

$$
\begin{aligned}
\|\mathbf{U}(\tau+1) - \mathbf{U}(0)\|_F &\le \sum_{s=0}^{\tau} \|\mathbf{U}(s+1) - \mathbf{U}(s)\|_2 \\
&= \sum_{s=0}^{\tau} \eta \|\nabla_{\mathbf{U}} \Phi(s)\|_F \\
&\le \sum_{s=0}^{\tau} \eta c_u \|\mathbf{X}\|_F \|\hat{\mathbf{y}}(s) - \mathbf{y}\|_2 \\
&\le \eta c_u \|\mathbf{X}\|_F \sum_{s=0}^{\tau} \left(1 - \eta \frac{\lambda_0}{2}\right)^{s/2} \|\hat{\mathbf{y}}(0) - \mathbf{y}\|_2 \\
&\le \frac{4}{\lambda_0} c_u \|\mathbf{X}\|_F \|\hat{\mathbf{y}}(0) - \mathbf{y}\|_2 \\
&\le \delta, \tag{225}
\end{aligned}
$$

$$
\begin{aligned}
\|\mathbf{a}(\tau+1) - \mathbf{a}(0)\|_F &\le \sum_{s=0}^{\tau} \|\mathbf{a}(s+1) - \mathbf{a}(s)\|_F \\
&= \sum_{s=0}^{\tau} \eta \|\nabla_{\mathbf{a}} \Phi(s)\|_F \\
&\le \eta \big(c_a \|\mathbf{X}\|_F + c_m\big) \sum_{s=0}^{\tau} \|\hat{\mathbf{y}}(s) - \mathbf{y}\|_2 \\
&\le \eta \big(c_a \|\mathbf{X}\|_F + c_m\big) \sum_{s=0}^{\tau} \left(1 - \eta \frac{\lambda_0}{2}\right)^{s/2} \|\hat{\mathbf{y}}(0) - \mathbf{y}\|_2 \\
&\le \frac{4}{\lambda_0} \big(c_a \|\mathbf{X}\|_F + c_m\big) \|\hat{\mathbf{y}}(0) - \mathbf{y}\|_2 \\
&\le \delta. \tag{226}
\end{aligned}
$$

Finally, using (197), we have

$$
\begin{aligned}
\|\mathbf{T}(\tau+1) - \mathbf{T}(0)\| &\le \frac{L}{1 - L\bar{\rho}_w} \big(c_a \|\mathbf{X}\|_F + c_m\big) \|\mathbf{W}(\tau+1) - \mathbf{W}(0)\|_2 \\
&\quad + \frac{L}{1 - L\bar{\rho}_w} \|\mathbf{U}(\tau+1) - \mathbf{U}(0)\|_2 \|\mathbf{X}\|_F \\
&\le \frac{L}{1 - L\bar{\rho}_w} \big(c_a \|\mathbf{X}\|_F + c_m\big) \frac{4}{\lambda_0} c_u \big(c_a \|\mathbf{X}\|_F + c_m\big) \|\hat{\mathbf{y}}(0) - \mathbf{y}\| \\
&\quad + \frac{L}{1 - L\bar{\rho}_w} \frac{4}{\lambda_0} c_u \|\mathbf{X}\|_F \|\hat{\mathbf{y}}(0) - \mathbf{y}\|_2 \|\mathbf{X}\|_F \\
&= \frac{4L}{(1 - L\bar{\rho}_w)\lambda_0} \left[ c_u \big(c_a \|\mathbf{X}\|_F + c_m\big)^2 + c_u \|\mathbf{X}\|_F^2 \right] \|\hat{\mathbf{y}}(0) - \mathbf{y}\|_2 \\
&\le \frac{2 - \sqrt{2}}{2} \sqrt{\lambda_0} \tag{227}
\end{aligned}
$$

by (6).

By Wely's inequality, it implies that the least singular value of $\mathbf{T}(\tau+1)$ satisfies $\sigma_{\min}(\mathbf{T}(\tau+1)) \ge \sqrt{\frac{\lambda_0}{2}}$. Thus, it holds $\lambda_{\tau+1} \ge \frac{\lambda_0}{2}$.

Now, we define $\mathbf{g} := \mathbf{a}(\tau+1)^T\mathbf{T}(\tau)$ and note that

$$
\Phi(\tau+1) - \Phi(\tau)
$$
$$
= \frac{1}{2}\|\hat{\mathbf{y}}(\tau+1) - \hat{\mathbf{y}}(\tau)\|_2^2 + (\hat{\mathbf{y}}(\tau+1) - \mathbf{g})^T(\hat{\mathbf{y}}(\tau) - \mathbf{y}) + (\mathbf{g} - \hat{\mathbf{y}}(\tau))^T(\hat{\mathbf{y}}(\tau) - \mathbf{y}). \tag{228}
$$

We bound each term of the RHS of this equation individually. First, using (198), we have

$$
\|\hat{\mathbf{y}}(\tau+1) - \hat{\mathbf{y}}(\tau)\|_2
$$
$$
\leq \bar{\rho}_a\left[\frac{L}{1 - L\bar{\rho}_w}\big(c_a\|\mathbf{X}\|_F + c_m\big)\|\mathbf{W}(\tau+1) - \mathbf{W}(\tau)\|_2\right.
$$
$$
\left. + \frac{L}{1 - L\bar{\rho}_w}\|\mathbf{U}(\tau+1) - \mathbf{U}(\tau)\|_2\|\mathbf{X}\|_F\right] + \big(c_a\|\mathbf{X}\|_F + c_m\big)\|\mathbf{a}(\tau+1) - \mathbf{a}(\tau)\|_2
$$
$$
= \bar{\rho}_a\left[\frac{L}{1 - L\bar{\rho}_w}\big(c_a\|\mathbf{X}\|_F + c_m\big)\eta c_u(c_a\|\mathbf{X}\|_F + c_m)\|\hat{\mathbf{y}}(\tau) - \mathbf{y}\|_2\right.
$$
$$
\left. + \frac{L}{1 - L\bar{\rho}_w}\eta c_u\|\mathbf{X}\|_F\|\hat{\mathbf{y}}(\tau) - \mathbf{y}\|_2\|\mathbf{X}\|_F\right]
$$
$$
+ \big(c_a\|\mathbf{X}\|_F + c_m\big)\eta(c_a\|\mathbf{X}\|_F + c_m)\|\hat{\mathbf{y}}(\tau) - \mathbf{y}\|_2
$$
$$
= \eta C_1\|\hat{\mathbf{y}}(\tau) - \mathbf{y}\|_2, \tag{229}
$$

where $C_1 := c_u^2(c_a\|\mathbf{X}\|_F + c_m)^2 + c_u^2\|\mathbf{X}\|_F^2 + (c_a\|\mathbf{X}\|_F + c_m)^2$.

On the other hand, we have

$$
(\hat{\mathbf{y}}(\tau+1) - \mathbf{g})^T(\hat{\mathbf{y}}(\tau) - \mathbf{y})
$$
$$
= \mathbf{a}(\tau+1)^T(\mathbf{T}(\tau+1) - \mathbf{T}(\tau))(\hat{\mathbf{y}}(\tau) - \mathbf{y})
$$
$$
\leq \|\mathbf{a}(\tau+1)\|_2\|\mathbf{T}(\tau+1) - \mathbf{T}(\tau)\|_2\|\hat{\mathbf{y}}(\tau) - \mathbf{y}\|_2
$$
$$
\leq \|\mathbf{a}(\tau+1)\|_2\left[\frac{L}{1 - L\bar{\rho}_w}\big(c_a\|\mathbf{X}\|_F + c_m\big)\|\mathbf{W}(\tau+1) - \mathbf{W}(\tau)\|_2\right.
$$
$$
\left. + \frac{L}{1 - L\bar{\rho}_w}\|\mathbf{U}(\tau+1) - \mathbf{U}(\tau)\|_2\|\mathbf{X}\|_F\right]\|\hat{\mathbf{y}}(\tau) - \mathbf{y}\|_2
$$
$$
\leq \bar{\rho}_a\left[\frac{L}{1 - L\bar{\rho}_w}\big(c_a\|\mathbf{X}\|_F + c_m\big)\|\eta c_u\big(c_a\|\mathbf{X}\|_F + c_m\big)\|\hat{\mathbf{y}}(\tau) - \mathbf{y}\|_2\right.
$$
$$
\left. + \frac{L}{1 - L\bar{\rho}_w}\eta c_u\|\mathbf{X}\|_F\|\hat{\mathbf{y}}(\tau) - \mathbf{y}\|_2\|\mathbf{X}\|_F\right]\|\hat{\mathbf{y}}(\tau) - \mathbf{y}\|_2
$$
$$
= \eta C_2\|\hat{\mathbf{y}}(\tau) - \mathbf{y}\|_2^2, \tag{230}
$$

where $C_2 := c_u^2\big(c_a\|\mathbf{X}\|_F + c_m\big)^2 + c_u^2\|\mathbf{X}\|_F^2$.

Furthermore, we also have

$$
(\mathbf{g} - \hat{\mathbf{y}}(\tau))^T(\hat{\mathbf{y}}(\tau) - \mathbf{y})
$$
$$
= (\mathbf{a}(\tau+1) - \mathbf{a}(\tau))^T\mathbf{T}(\tau)(\hat{\mathbf{y}}(\tau) - \mathbf{y})
$$
$$
= -\big(\eta\nabla_{\mathbf{a}}\Phi(\tau)\big)^T\mathbf{T}(\tau)(\hat{\mathbf{y}}(\tau) - \mathbf{y})
$$
$$
= -\eta(\hat{\mathbf{y}}(\tau) - \mathbf{y})^T\mathbf{T}(\tau)^T\mathbf{T}(\tau)(\hat{\mathbf{y}}(\tau) - \mathbf{y})
$$
$$
\leq -\eta\frac{\lambda_0}{2}\|\hat{\mathbf{y}}(\tau) - \mathbf{y}\|_2^2 \tag{231}
$$

where we use induction $\lambda_\tau \geq \frac{\lambda_0}{2}$.

From (228)-(231), we obtain

$$
\begin{aligned}
\Phi(\tau+1) &- \Phi(\tau) \\
&\leq \frac{1}{2}\eta^2 C_1^2 \|\hat{\mathbf{y}}(\tau) - \mathbf{y}\|_2^2 + \eta C_2 \|\hat{\mathbf{y}}(\tau) - \mathbf{y}\|_2^2 - \eta\frac{\lambda_0}{2}\|\hat{\mathbf{y}}(\tau) - \mathbf{y}\|_2^2 \\
&= 2\Phi(\tau)\left[\frac{1}{2}\eta^2 C_1^2 + \eta C_2 - \eta\frac{\lambda_0}{2}\right] \\
&= \Phi(\tau)\left[\eta^2 C_1^2 + 2\eta C_2 - \eta\lambda_0\right],
\end{aligned}
$$

which leads to

$$
\begin{aligned}
\Phi(\tau+1) &\leq \Phi(\tau)\left[1 - \eta(\lambda_0 - \eta C_1^2 - 2C_2)\right] \\
&\leq \big(1 - \eta(\lambda_0 - 4C_2)\big)\Phi(\tau) \\
&\leq \left(1 - \eta\frac{\lambda_0}{2}\right)\Phi(\tau) \tag{232} \\
&\leq \left(1 - \eta\frac{\lambda_0}{2}\right)^{\tau+1}\Phi(0), \tag{233}
\end{aligned}
$$

where (233) follows from the induction assumption (220). This concludes our proof of (218)–(220) by induction.

Finally, (8) is a direct application of (Ling et al., 2022, Eq. (5)), using the fact that $\lambda_\tau \geq \lambda_0/2$ as stated in (219).

## H  Proof of Lemma 19

Observe that

$$
\|\hat{\mathbf{y}}(0) - \mathbf{y}\| \leq \|\hat{\mathbf{y}}(0)\| + \|\mathbf{y}\|. \tag{234}
$$

On the other hand, let $\mathbf{v}_1, \mathbf{v}_2, \cdots, \mathbf{v}_n$ be $n$ columns of $\mathbf{T}(0)$. Then, we have

$$
\begin{aligned}
\|\hat{\mathbf{y}}(0)\| &= \left\|\mathbf{a}^T(0)\mathbf{T}^*(0)\right\| \\
&= \sqrt{\sum_{i=1}^n (\mathbf{a}^T(0)\mathbf{v}_i)^2}. \tag{235}
\end{aligned}
$$

Now, observe that

$$
\begin{aligned}
\mathbb{P}\bigg[\sum_{i=1}^n (\mathbf{a}^T(0)\mathbf{v}_i)^2 &\geq \frac{n}{t}\left(\frac{8L^2 d + 2mL^2}{m(1 - 8L^2\sigma_w^2)}\right)\bigg] \\
&\leq \frac{t}{n}\left(\frac{m(1 - 8L^2\sigma_w^2)}{8L^2 d + 2mL^2}\right)\mathbb{E}\bigg[\sum_{i=1}^n (\mathbf{a}^T(0)\mathbf{v}_i)^2\bigg] \\
&= \frac{t}{n}\left(\frac{m(1 - 8L^2\sigma_w^2)}{8L^2 d + 2mL^2}\right)\sum_{i=1}^n \mathbb{E}\big[(\mathbf{a}^T(0)\mathbf{v}_i)^2\big] \\
&= \frac{t}{n}\left(\frac{m(1 - 8L^2\sigma_w^2)}{8L^2 d + 2mL^2}\right)\sum_{i=1}^n \frac{\mathbb{E}[\|\mathbf{v}_i\|^2]}{m} \tag{236} \\
&\leq \frac{t}{n}\left(\frac{m(1 - 8L^2\sigma_w^2)}{8L^2 d + 2mL^2}\right)\sum_{i=1}^n \frac{1}{m}\left(\frac{8L^2 d + 2mL^2}{1 - 8L^2\sigma_w^2}\right) \tag{237} \\
&= t, \tag{238}
\end{aligned}
$$

where (236) follows from Assumption 1 that $\mathbf{a}$ is initialised with a random vector with i.i.d. entries $\mathcal{N}(0, 1/m)$ and the fact that $\mathbf{a}(0)$ is independent of $\mathbf{v}_i$, and (237) follows from (177).

From (238), with probability at least $1 - t$ it holds that

$$\sum_{i=1}^{n} (\mathbf{a}^T(0)\mathbf{v}_i)^2 \leq \frac{n}{t}\left(\frac{8L^2 d + 2mL^2}{m(1 - 8L^2\sigma_w^2)}\right) = O(n),$$

which leads to

$$\|\hat{\mathbf{y}}(0)\| = O(\sqrt{n}) \tag{239}$$

by (235).

In addition, we have

$$\|\mathbf{y}\| = \sqrt{\sum_{i=1}^{n} y_i^2}$$
$$= O(\sqrt{n}) \tag{240}$$

by Assumption 2.

From (234), (239), and (240) we obtain

$$\|\hat{\mathbf{y}}(0) - \mathbf{y}\| = O(\sqrt{n}). \tag{241}$$

# I Proof of Theorem 20

By using Theorem 7 and Theorem 8, it holds that $\lambda_* \geq \lambda_0^* > 0$ where $\lambda_0^*$ is a function of $n$ only, which does not depend on $m$. On the other hand, by Lemmas 18, it holds with probability $1 - \exp(-\Omega(m))$ that

$$\bar{\rho}_w = O(1), \qquad \bar{\rho}_u = O(1), \qquad \bar{\rho}_a = O(1),$$

which implies that

$$c_a = O(1), \qquad c_u = O(1), \qquad c_m = 0.$$

Now, by Theorem 7 and Theorem 8, it holds with probability $1 - t$ that

$$0 < \lambda_* \leq \frac{2}{m}\lambda_0 \leq \frac{2}{m}\mathrm{tr}(\mathbf{G})$$
$$\leq \frac{2}{m}\sum_{i=1}^{n}\left(\frac{L\sqrt{d} + \sqrt{m}L}{1 - 2L\sqrt{2}\sigma_w}\right)^2 = O(n), \tag{242}$$

where (242) follows from (75).
In addition, by Assumption 2 we have

$$\|\mathbf{X}\|_F = \sqrt{nd}. \tag{243}$$

Hence, by combining with Lemma 19, i.e., $\|\hat{\mathbf{y}}(0) - \mathbf{y}\| = O(\sqrt{n})$, the inequalities (37)-(39) hold with probability at least $1 - t$ if $m = \Omega\left(\frac{n^3}{(\lambda_*)^2}\log\frac{n}{t}\right)$.

