# OpenReview forum: "Global Convergence Rate of Deep Equilibrium Models with General Activations"
_TMLR — Accepted by TMLR_

### Review · Reviewer_PWEb · 2024-11-02

**Summary Of Contributions:**

This paper investigates the global convergence rate of deep equilibrium models with general activation functions. Here, a ``general activation function'' refers to functions with bounded first and second derivatives. A Gram matrix is constructed, and several theorems regarding its eigenvalues are presented. Additionally, examples are provided to demonstrate that Theorem 6 holds for certain activation functions.

**Audience:**

Yes

**Broader Impact Concerns:**

Not available.

**Claims And Evidence:**

Yes

**Requested Changes:**

1. Page 2, Typo: Correct the phrase "In many many existing deep sequence models" to "In many existing deep sequence models."
2. Weight Initialization Algorithm: It would be helpful to provide an algorithm for initializing weights that satisfy the assumptions in Theorem 2. Consider detailing a method that aligns with these assumptions to facilitate practical implementation.
3. Clarification of Notation: The notation $\nparallel$ is unclear and would benefit from a precise definition. Additionally, it may be worthwhile to discuss the effect of repeated input data, particularly regarding whether such repetitions impact the validity of the current results.

**Strengths And Weaknesses:**

## Strengths:
This paper provides insights into scaling up deep equilibrium models by analyzing their convergence properties.

## Weaknesses:
1. The analysis based on an infinite-layer model may oversimplify the actual model. Although the limit is well-defined and has an appropriate convergence rate, a key question remains: will results from the finite-layer case converge to this behavior, and is this asymptotic behavior beneficial in real-world scenarios?
2. Theorem 1's statement presents an issue: the eigenvalue bound is influenced by data and model specifics, making it challenging to properly initialize weights.
3. It’s difficult to extract a quantitative assessment from this work. A valuable addition would be exploring how the finite-layer model's behavior relates to the infinite-layer scenario.
4. For Theorems 5 and 6, the broader implications for the practical use of deep equilibrium models remain unclear, making it hard to gauge their significance to the larger community working with these models.

---

> ### Author Response · Authors · 2024-11-27
>
> Thank you very much for your comments on our paper. The following are our answers to your changed requests:
>
> 1. Weight Initialization Algorithm: It would be helpful to provide an algorithm for initializing weights that satisfy the assumptions in Theorem 2. Consider detailing a method that aligns with these assumptions to facilitate practical implementation.
>
> Answer: Based on your suggestion, we have added Section 7 in the revised version, where we propose an algorithm to initialize weights that satisfy the assumptions in Theorem 2. Additionally, we provide a proof in Theorem 19 of this section, demonstrating why the algorithm will terminate with high probability.
>
> 3. Clarification of Notation: The notation $\nparallel $
>  is unclear and would benefit from a precise definition. Additionally, it may be worthwhile to discuss the effect of repeated input data, particularly regarding whether such repetitions impact the validity of the current results.
>
> Answer: In Definition 2 in the revised version, we provide a definition of $\nparallel$. We also discuss the effect of repeated input data right after (1). More specifically, we add the following:
>
> "If we were to repeat this update an infinite number of times, we would essentially be modeling an infinitely deep network of the form above.  In practice, what we find is that for most “typical” deep layers the valued actually converge to a fixed point or equilibrium point \citep{Bai2019DeepEM}.  The output of the last hidden layer is defined by $\mathbf{T}^*:=\lim\_{l\to \infty} \mathbf{T}^{(l)}$."
>
> Shaojie Bai, J. Zico Kolter, Vladlen Koltun, "Deep Equilibrium Models".

---

### Review · Reviewer_5MPc · 2024-11-18

**Summary Of Contributions:**

This paper studies deep equilibrium models with activation functions having bounded first and second-order derivatives. Under certain conditions on the initialization, it is shown that gradient descent converges to a globally optimal solution at a linear rate.

**Audience:**

Yes

**Claims And Evidence:**

Yes

**Requested Changes:**

Requested changes:

* The main novelty in the paper is creating a new population Gram matrix in Definition 4. Could the authors provide some insight into how they chose that particular form of the Gram matrix? Such insights may better help understand the Gram matrix.

* The paper contains numbered equations that are not referenced in the text. Removing numbering for unused equations would enhance the readability and reduce unnecessary clutter.

**Strengths And Weaknesses:**

Strengths:

* Compared to previous work [1], this work establishes convergence of gradient descent for a broader class of activation functions. This is accomplished by creating a new population Gram matrix.

Weaknesses:

*  The main weakness is the scope of this paper, as it extends the result of ReLU activation function proved in [1] to a more general activation function. Apart from introducing a new population Gram matrix, the rest of the proof follows a similar approach as [1].

* At the beginning of the proof of **Theorem 6**, the authors argue that $Q_{ij}(x)$ has the same form as $\tilde{Q}\_\{\alpha,\alpha\}(x)$ for some $\alpha \geq 1$, because $E[G_{ii}] = E[G_{jj}]$ for all $i, j$. However, $Q_{ij}(x)$ depends on $G_{ii}$ and $G_{jj}$, instead of $E[G_{ii}]$ and $E[G_{jj}]$. I am not sure how it follows that $Q_{ij}(x)$ has the same form as $\tilde{Q}_{\alpha,\alpha}(x)$. Could the authors please explain this?

[1] Zenan Ling, Xingyu Xie, Qiuhao Wang, Zongpeng Zhang, and Zhouchen Lin. Global convergence of over-parameterized deep equilibrium models.

---

> ### Author Response · Authors · 2024-11-27
>
> Thank you very much for your comments on our paper.  The following are our answers to your questions:
>
> 1. At the beginning of the proof of Theorem 6, the authors argue that $Q_{ij}(x)$ has the same form as $\tilde{Q}\_{\alpha,\alpha}$ for some $\alpha \geq 1$ because $\mathbb{E}[G\_{ii}]=\mathbb{E}[G\_{jj}]$ for all $i,j$. However, $Q_{ij}(x)$ depends on $G_{ii}$ and $G_{jj}$, instead of  $\mathbb{E}[G\_{ii}]$ and $\mathbb{E}[G\_{jj}]$. I am not sure how it follows that $Q_{ij}(x)$ has the same form as $\tilde{Q}_{\alpha,\alpha}$. Could the authors please explain this?
>
> Answer:  This is a mistake (typo) in our statement of this theorem. The correct definition of $Q_{ij}(x)$ in Definition 4 (old version) (or Definition 5 in our revised version) is
> $$
> Q\_{ij}(x):=\tilde{Q}\_{\sqrt{2\big(\frac{\sigma_w^2}{m}\mathbb{E}[\mathbf{G}\_{ii}]+1\big)},\sqrt{2\big(\frac{\sigma_w^2}{m}\mathbb{E}[\mathbf{G}\_{jj}]+1\big)} }\big(x\big), \qquad \forall x \in \mathbb{R}.
> $$
>
> 2.  The main novelty in the paper is creating a new population Gram matrix in Definition 4. Could the authors provide some insight into how they chose that particular form of the Gram matrix? Such insights may better help understand the Gram matrix.
>
> Answer: We understand your points. However, our design of this population Gram matrix is totally based on mathematical reasoning. It is hard to say why we come to this matrix. But, we aim to design a sequence of matrices $\mathbf{K}^{(l)}$ such that $\bigg\\|\frac{1}{m}\mathbf{G}^{(l)}-\mathbf{K}^{(l)}\bigg\\|\_F=O\bigg(n(2L\sqrt{2} \sigma\_w)^l\bigg)$ and $\mathbf{K}^{(l)}\to \mathbf{K}$ which is a  positive definite matrix.  It is partly based on our observation that $\hat{\mathbf{G}}\_{ij}^{(l)}$ has the form (149) in the old version (or (94) in the revised version).
>
> 3. The paper contains numbered equations that are not referenced in the text. Removing numbering for unused equations would enhance the readability and reduce unnecessary clutter.
>
> Answer: Thank you for your suggestion. We have removed nearly all of the numbered equations that were not referenced in the text. The previous version contained 354 equations, while the revised version now includes only 218.

---

### Review · Reviewer_ZV3G · 2024-12-10

**Summary Of Contributions:**

This paper extends prior work on the convergence of Deep Equilibrium Models, that relied on ReLU activations, to new non-linear functions with bounded first- and second-order derivatives. The challenge in extending the existing work relies in the construction of a “population” Gram matrix and the use of Hermite polynomial expansions. Numerical results are evaluated on MNIST and CIFAR-10 to validate the convergence result of their theoretical analysis.

**Audience:**

Yes

**Broader Impact Concerns:**

No broader impact concerns.

**Claims And Evidence:**

Yes

**Requested Changes:**

**Further comments**

I had some further comments and requested changes:

- Equation (4) is very abruptly introduced. It would be helpful to have some context when it is first introduced: intuitively what does it imply and why is it important?
- The takeaways of theorem 2 are confusing/not directly following from the results. Specifically:
    - The fact that $\lVert W(\tau)\rVert_2 \leq 1$ does not mean the equilibrium point ($T^{\star}$) exists. This also depends on $U(\tau)$, but also, regardless of that, a bounded norm could imply oscillatory behavior on the weights.
    - (10) is intuitively confusing. If the gradient of the loss is always larger than the loss value, doesn’t that indicate that convergence to a zero (or constant) loss might not happen? I’m assuming the author’s intend to argue that a reduction in loss will occur, however if the optimal loss value is not zero, then this indicates divergence.
- Somewhat minor, but below (19) it says that the $\lambda_0$ can be bound by the least eigenvalue of $K$, but $K$ is not defined. How is $K$ related to $K^{(l)}$?
- Section 5 is very difficult to parse. There are back-to-back theorems and propositions with no intuitive explanations and it is not clear what we’re working towards/how these intermediate results build up to the proof of Theorem 5.
- For Section 6, it is not clear from the exposition why Theorem 6 is needed. The motivation was that designing $K$ was the challenge, and that seems to be complete in Section 5. How does Theorem 6 fit in the overall setting of the paper?
- The section above Example 13 is very confusing. It defines a new activation function, $\hat{\phi}$, and then it says that this new activation function satisfies (42), if it satisfies $\hat{\phi}(x) = \sum_{n=1}^{\infty}a^2_nx^n$. This seems recursive, why not assume (42) directly?
- What is the takeaway of the second experiment? No discussion was made about how different activation functions could affect the dynamics, and this result is just shown with no follow-up discussion.
- The conclusion (and different sections in the manuscript) talk about globally optimal solutions, however I’m unsure where that is introduced. To my understanding, Theorem 2 is about convergence. Where does the optimality come in?
- Finally, you mention extensions to other classes of activation functions. What are some examples of such classes that are not covered with your framework? It would be interesting to include some of these in your experiments and see if they diverge, or if they exhibit similar dynamics.

**Typos**

First page, 7 lines from the bottom: “on generalization error of convex combination” → “on the generalization error of convex combinations”

First page, 2 lines from the bottom: “minimise a generalisation” UK English, rest of the manuscript uses US English.

Second page, first line: “information theoretic approach” → “information theoretic approaches”

Second page, fourth line: “deep equilibrium model” → “deep equilibrium models”

Second page, second paragraph: “in many research literature” → don’t know exactly how to fix this, grammatical errors

Second page, third paragraph: “have homogeneous property” → “have the homogeneous property”

Section 2: “Problem settings” → “Problem setting”

Above (1): “with the transform of the l-th layer as” → “with the l-th layer transforming as”

Below (4): “Besides, we use” → Use of “besides” doesn’t seem appropriate

Theorem 2: No norm is specified for W(0). What is $\sigma(0)$?

Below (11): “causes … can not be applied” → non-legible

Last paragraph before Section 3: “with general Lipschitz activation function” → “with general Lipschitz activation functions”

Definition 4: The word “Given” doesn’t seem appropriate.

Above (41): “associated with” → “associated with”

Section 7: “we variate” → “we vary”

Section 7: “training dynamic” → “training dynamics”

Page 9: “we variate” → “we vary”

Conclusion: “CFAR” → ”CIFAR”

**Strengths And Weaknesses:**

**Strengths**

The extension of the prior work to activation functions beyond ReLU has potential to be useful in certain application domains that necessitate the use of different activation functions. The analyses seem thorough and draw results from many fields, which can be challenging.

**Weaknesses**

The main weaknesses of the work revolve around motivation, presentation, and the experimental section.

*Motivation*

The authors mention that it is an interesting question whether DEQs converge with non-linear functions. Besides the fact that ReLU is itself a non-linear function, it is not motivated why this is an interesting question. ReLU has many added benefits, such as sparsity, low-energy usage, etc, and it is not clear why the authors would like to use other activation functions.

*Presentation*

The presentation of the work is not currently conducive to understanding. A high-level overview of the results is missing and how they connect to one another can be easily lost on the reader. Most results are stated, one after another, without placing them in the overall context of the proof procedure or highlighting their importance/presenting them in plain words. Examples of this include (6-8), which are hard to parse and no intuitive explanation is given. Similarly, the term “population Gram matrix” is not defined in the manuscript, even though it is used extensively, and it is not explained how it relates to the problem and why statements about this matrix translate to statement about the original problem. Finally, there is a large number of typos and grammatical errors (I will list them, alongside other presentation-related issues, in the next section).

*Experiments*

The experimental section is weak, even for a theory paper. However, the biggest weakness seems to be the dissonance between the experimental results and the developed theory. Specifically, when first introduced in the introduction, DEQs are motivated as models for modeling sequences. However, neither MNIST or CIFAR-10 are sequential data, so why where they chosen as the datasets for the experiments? Also the choice to include only two classes and 500 samples (when MNIST and CIFAR-10 are relatively “easy” datasets) hurts the credibility of the section. As another comment, the section is further confusing as DEQs are introduced as computing the equilibrium point directly, and thus avoiding the iterative computation. For the experiments, it is stated that a DEQ is used, but then iterative computation is presented.

---

> ### Author Response · Authors · 2024-12-19
>
> Thank you for your valuable comments and helpful suggestions for improving our paper. Below, we provide our responses to your feedback, starting with the identified weaknesses.
>
> 1. Motivation:
>
> In the revised version, we have included a paragraph on page 2, Section 1, explaining our rationale for exploring alternative activation functions, as follows:
>
> *It is crucial to explore alternative activation functions in DEQs because ReLU can lead to issues like dead neurons, gradient saturation, and instability in fixed-point iterations. Functions such as GELU, Swish, and tanh provide smoother gradients, better gradient flow, and more expressive non-linearities, which contribute to improved stability, faster convergence, and enhanced generalization. These alternatives address the limitations of ReLU, potentially improving model performance and making DEQs more effective at capturing complex patterns, especially in tasks that demand stable and efficient training.*
>
> 2. Presentation
>
> In the revised version, we have made every effort to improve the presentation based on your suggestions. Please find our responses to your requested changes in the following section. However, due to the highly technical nature of this work, providing intuitive explanations for all the formulas is quite challenging. Regarding your comment about the definition of the population Gram matrix, we have added the following definition in Section 3:
>
> *The main challenge now is to find an appropriate initialization such that $\lambda_0$  satisfies all the conditions in Theorem 3. Estimating $\lambda_0$
>  directly is difficult, and a common strategy is to establish a concentration inequality between *the initial empirical Gram matrix* $\mathbf{G}$ and a new matrix with a more easily estimable least eigenvalue. This new matrix is referred to as *the population Gram matrix* and is denoted by
> $\mathbf{K}$ \cite{Ling2022GlobalCO}. However, due to the non-homogeneity of the new activation function $\varphi$, bounding $\lambda_0$
>  becomes more challenging than in the case of ReLU networks, as discussed in \cite{Ling2022GlobalCO}. The non-homogeneity of the activation functions makes the design techniques for $\mathbf{K}$ presented in \cite[Definition 1]{Ling2022GlobalCO} inapplicable. For example, \cite[Eq.~11]{Ling2022GlobalCO} is only valid for the ReLU function.*
>
> 3. Experiments
>
> MNIST and CIFAR-10 were selected not because they are sequential datasets, but because they offer a well-established and straightforward environment to validate the linear convergence rate of the square loss in DEQs. In response to your concerns, we have implemented DEQs on the full MNIST and CIFAR-10 datasets in this revised version. In our experiments, we indeed found a fixed-point $\mathbf{T}$ of the equation $\mathbf{T} = \varphi(\mathbf{W} \mathbf{T} + \mathbf{U} \mathbf{X})$ using the Anderson acceleration method.
>
> Additionally, we intended to conduct experiments on a sequential dataset, as per your suggestion. However, due to time constraints, we were unable to complete this task. The running and testing times for such datasets are quite long, which limited our ability to carry out these experiments within the available time frame. We plan to include these experiments in a future phase of the work.

---

> > ### Author Response · Authors · 2024-12-19
> >
> > In the following section, we provide responses to the changes you requested:
> >
> > + *Equation (4) is very abruptly introduced. It would be helpful to have some context when it is first introduced: intuitively what does it imply and why is it important?*
> >
> > Answer: $q$ is simply a normalized constant in the definition of $\tilde{Q}_{\alpha,\beta}$ in Definition 5. In the revised version, we have moved the definition of $q$ to Definition 5 for clarity.
> > + *The takeaways of theorem 2 are confusing/not directly following from the results. Specifically:*
> >    + *The fact that $\\|\mathbf{W}(\tau)\\|_2<1$ does not does not mean the equilibrium point* ($\mathbf{T}^*$) *exists. This also depends on $U(\tau)$, but also, regardless of that, a bounded norm could imply oscillatory behavior on the weights.*
> >
> > Answer: In the revised version, we have provided a proof for the existence of the equilibrium point $(\mathbf{T}^*)$ under the condition $\\|\mathbf{W}(\tau)\\|_2<1/L$ (apologies for the typo in the previous version) in Lemma 25. The essential idea is that, under the assumption $\\|\mathbf{W}(\tau)\\|_2<1/L$, the sequence $\\{\mathbf{T}^{(l)}\\}\_{l=1}^{\infty}$ forms a Cauchy sequence in the Banach space $\mathbf{R}^{m\times n}$. As a result, this sequence must converge to a fixed point $\mathbf{T}^*$.
> >
> >  + *(10) is intuitively confusing. If the gradient of the loss is always larger than the loss value, doesn’t that indicate that convergence to a zero (or constant) loss might not happen? I’m assuming the author’s intend to argue that a reduction in loss will occur, however if the optimal loss value is not zero, then this indicates divergence.*
> >
> > Answer: Equation (10) represents the Polyak-Lojasiewicz (PL) condition, which is widely used to establish the linear convergence rate of gradient descent (GD). As shown in equation (9) of Theorem 3 (in the revised version), the loss converges to zero as $\tau \to \infty$, due to the choice of the convergence rate $\eta< 2/\lambda_0$.
> >
> > + *Somewhat minor, but below (19) it says that the $\lambda_0$ can be bound by the least eigenvalue of $\mathbf{K}$, but $\mathbf{K}$ is not defined. How $\mathbf{K}$ is related to $\mathbf{K}^{(l)}$?*
> >
> > Answer:  In our revised version, we have provided a definition for $\mathbf{K}$ right before Theorem 7, as follows:
> >
> > "The next result shows that $\lambda_0$ can be lower-bounded via the least eigenvalue of the population matrix $\mathbf{K}$, where $\mathbf{K}=\lim_{l\to \infty} \mathbf{K}^{(l)}$. The existence of this limit will be proved in Proposition 13.
> >
> > + *Section 5 is very difficult to parse. There are back-to-back theorems and propositions with no intuitive explanations and it is not clear what we’re working towards/how these intermediate results build up to the proof of Theorem 5.*
> >
> > Answer: As mentioned in our response to the weaknesses section, due to the technical nature of this work, providing intuitive explanations for all of the formulas is challenging. However, in order to address the concern you raised, we have made efforts to clarify the connections between our results. Specifically, we now explicitly state that Lemma 9, Lemma 10, and Lemma 11 are used to prove Propositions 12, 13, and 14. These propositions, in turn, form the foundation for proving Theorem 7.
> >
> > + *For Section 6, it is not clear from the exposition why Theorem 6 is needed. The motivation was that designing $\mathbf{K}$ was the challenge, and that seems to be complete in Section 5. How does Theorem 6 fit in the overall setting of the paper?*
> >
> > Answer: To clarify the necessity of Theorem 6 (or Theorem 8 in the revised version), we have added the following paragraph between Theorem 7 and Theorem 8 in this revised version:
> >
> > "Based on Theorem 7, we can show that there exists a weight initialization algorithm (WIAL) such that all the conditions in (5)--(7) of Theorem 3 are satisfied for sufficiently large $m$, provided that
> > $\lambda_*>0$ (cf. Section 7). The following result establishes a sufficient condition for
> > $\lambda_* >0$, or equivalently, for $\mathbf{K}$ to be strictly positive definite."
> >
> > + *The section above Example 13 is very confusing. It defines a new activation function, $\hat{\varphi}$, and then it says that this new activation function satisfies (42), if it satisfies $\hat{\varphi}(x)=\sum_{n=1}^{\infty} a_n^2 x^n$. This seems recursive, why not assume (42) directly?*
> >
> > Answer: The expression for $\tilde{Q}\_{\alpha,\alpha}$ in (42) depends on the Hermite expansion of $\varphi(x)$. Since $\varphi(x)$ and $\hat{\varphi}(x)$ are related, we can use either $\varphi(x)$ or $\hat{\varphi}(x)$ to derive $\tilde{Q}_{\alpha,\alpha}$. To avoid confusion, we have rewritten this explanation at the beginning of Section 6 in our revised version.

---

> > > ### Author Response · Authors · 2024-12-19
> > >
> > > + *What is the takeaway of the second experiment? No discussion was made about how different activation functions could affect the dynamics, and this result is just shown with no follow-up discussion.*
> > >
> > > Answer: We conducted the second experiment to demonstrate that the linear convergence rate of the square loss holds for DEQs with different activation functions. However, we acknowledge that our original statement was not clear enough. In this revised version, we have added a clearer takeaway from this experiment, following your suggestion (please refer to our answer to the last question).
> > >
> > > +  *The conclusion (and different sections in the manuscript) talk about globally optimal solutions, however I’m unsure where that is introduced. To my understanding, Theorem 2 is about convergence. Where does the optimality come in?*
> > >
> > > Answer: As we can observe from the main theorem (Theorem 3 in the revised version or Theorem 2 in the previous version), $\Phi(\theta(\tau))\leq \big(1-\eta \frac{\lambda_0}{2}\big)^{\tau} \Phi(\theta(0))\to 0$ as $\tau\to \infty$ since we chose $\eta< 2/\lambda_0$. This implies that $\\|\hat{\mathbf{y}}(\theta(\tau))-\mathbf{y}\\|_2 \to 0$ (cf. (3) in our revised version), or equivalently $\hat{\mathbf{y}}(\theta(\tau)) \to \mathbf{y}$. This is where optimality comes into play. In response to your concerns, we have made this point clearer in Theorem 3 of the revised version.
> > >
> > > + *Finally, you mention extensions to other classes of activation functions. What are some examples of such classes that are not covered with your framework? It would be interesting to include some of these in your experiments and see if they diverge, or if they exhibit similar dynamics.*
> > >
> > > Answer: Based on this insightful suggestion, we have conducted additional experiments with DEQs using various activation functions and derived key takeaways from our findings.
> > >
> > > "In the second experiment, we vary the activation function and plot the training dynamics for MNIST and CIFAR-10 with $m=3000$
> > > for several activation functions, including ReLU, Sin, Identity, Swish, and Mish. While the activation functions Swish and Mish do not have bounded first derivatives and are thus not
> > > $L$-bounded, we observe from Fig. 2 that when $m$ is sufficiently large and $\tau$ is sufficiently large, all the networks converge to the global optimum at a linear rate. This suggests that the $L$-bounded condition is a sufficient, but not necessary, condition for the linear convergence rate of the squared loss."
> > >
> > > >> Finally, we would like to thank the reviewers for pointing out grammar errors and typos in our manuscript. We have corrected these issues and have also carefully re-read the manuscript to identify and address any remaining errors.

---

### Decision · Action_Editor_8AdW · 2025-02-03

**Recommendation:** Accept as is

**Comment:**

Reviewers felt that theory established in this paper lies on solid footing, but had concerns regarding motivation (importance of activation functions beyond ReLU), breadth of theoretical novelty, presentation and strength of empirical support.  The authors partially addressed these concerns.  While the paper is in my opinion still limited, particularly in its significance, such limitations should not be an impediment for publication in TMLR.

**Audience:**

There will likely be at least some individuals within TMLR readership that will be interested in the contribution

**Claims And Evidence:**

Theoretical and empirical claims made in the paper are properly supported by evidence